JCB Journal of Cell Biology

# The role of the AP-1 adaptor complex in outgoing and incoming membrane traffic

Margaret S. Robinson[1], Robin Antrobus[1], Anneri Sanger[1], Alexandra K. Davies[2], and David C. Gershlick[1]

The AP-1 adaptor complex is found in all eukaryotes, but it has been implicated in different pathways in different organisms. To look directly at AP-1 function, we generated stably transduced HeLa cells coexpressing tagged AP-1 and various tagged membrane proteins. Live cell imaging showed that AP-1 is recruited onto tubular carriers trafficking from the Golgi apparatus to the plasma membrane, as well as onto transferrin-containing early/recycling endosomes. Analysis of single AP-1 vesicles showed that they are a heterogeneous population, which starts to sequester cargo 30 min after exit from the ER. Vesicle capture showed that AP-1 vesicles contain transmembrane proteins found at the TGN and early/recycling endosomes, as well as lysosomal hydrolases, but very little of the anterograde adaptor GGA2. Together, our results support a model in which AP-1 retrieves proteins from post-Golgi compartments back to the TGN, analogous to COPI's role in the early secretory pathway. We propose that this is the function of AP-1 in all eukaryotes.

## Introduction

Adaptor protein (AP) complexes are an ancient family of heterotetramers, which select cargo for packaging into transport vesicles at various locations in the cell. The first AP complexes to be discovered, AP-1 and AP-2, were identified as major components of clathrin-coated vesicles (CCVs), with AP-1 acting at intracellular membranes and AP-2 at the plasma membrane. The three additional AP complexes were found by searching for homologs of AP-1 and AP-2. Out of the five complexes, AP-1 is arguably the most indispensable. It is the only one that is present in every eukaryote whose genome has been sequenced, and knocking it out in animals is invariably embryonic lethal (Robinson, 2015). AP-1 has several medical connections, including genetic disorders caused by mutations in some of the AP-1 subunits (Dell'Angelica and Bonifacino, 2019; Sanger et al., 2019) and hijacking of the complex by pathogens, most notably HIV (Lubben et al., 2007; Roeth et al., 2004). But in spite of AP-1's importance, its function is not well understood.

Initially, AP-1 was assumed to be involved in the receptor-mediated trafficking of newly synthesized lysosomal hydrolases, packaging them into CCVs at the trans-Golgi network (TGN). By using the clathrin/AP-1 pathway, the hydrolases would be diverted away from the secretory pathway and targeted to lysosomes via endosomes. However, the evidence for this has always been somewhat circumstantial. Early cytochemical studies by Novikoff and coworkers showed that lysosomal acid phosphatase (LAP) could be detected in clathrin-coated budding profiles near the Golgi apparatus (Holtzman et al., 1967). However, we now know that LAP is a transmembrane protein rather than a soluble hydrolase and that it is transported to lysosomes mainly via the plasma membrane (Braun et al., 1989). Thus, the LAP in the clathrin-coated profiles could have come from endosomes rather than the Golgi apparatus.

Subsequent studies showed that in vertebrates, most soluble hydrolases are trafficked by the cation-dependent and cation-independent mannose 6-phosphate (M6P) receptors, CDMPR and CIMPR. These receptors bind to newly synthesized hydrolases as they leave the Golgi and deliver them to endosomes, after which the empty receptors return to the TGN for another round. The two receptors can also traffic to the plasma membrane (Dahms et al., 1989). Both CDMPR and CIMPR have been detected in the preparations of isolated CCVs, and both are able to bind to AP-1 and AP-2 in vitro (Robinson, 2015). In addition, knockdowns and knockouts of AP-1 cause increased secretion of the hydrolase cathepsin D (Meyer et al., 2000; Hirst et al., 2003).

All of these studies are consistent with a role for AP-1 in the anterograde trafficking of M6P receptors and hydrolases, but they are equally consistent with a role for AP-1 in the retrieval of M6P receptors. Indeed, when AP-1 is depleted, the receptors are largely lost from the Golgi region and relocated to peripheral endosomes (Meyer et al., 2000, 2001; Robinson et al., 2010), supporting a role for AP-1 in the recycling of receptors back to the TGN, rather than in anterograde trafficking. The situation is

[1]Cambridge Institute for Medical Research, University of Cambridge, Cambridge, UK; [2]Faculty of Biology, Medicine and Health, School of Biological Sciences, Manchester Academic Health Science Centre, University of Manchester, Manchester, UK.

Correspondence to Margaret S. Robinson: msr12@cam.ac.uk.



further complicated by the presence of other types of machinery, such as GGAs, proteins found in opsithokonts (e.g., animals and fungi) that act together with clathrin to facilitate the forward trafficking of receptor-bound hydrolases (Boman, 2001). In yeast, knocking out AP-1 affects the retrieval of late Golgi resident proteins from endosomes, with no apparent effects on outward traffic (Valdivia et al., 2002), while knocking out GGAs impairs the transport of hydrolases from the late Golgi to endosomes (Hirst et al., 2000; Dell'Angelica et al., 2000; Costaguta et al., 2001). However, the relationship between AP-1 and GGAs in yeast is not entirely clear. Although they show little colocalization with each other (Daboussi et al., 2012), the combined knockout has a more severe phenotype than either knockout alone (Hirst et al., 2000; Costaguta et al., 2001).

Knocking out or knocking down AP-1 in animal cells has a surprisingly subtle phenotype. Although hydrolase receptors and other late Golgi proteins are relocated to peripheral endosomes, most are still present in normal amounts in isolated CCV-enriched fractions (Hirst et al., 2012; Navarro Negredo et al., 2017). Because of the possibility that cells might be able to compensate for the gradual loss of AP-1, we developed the knocksideways system, which rapidly depletes the available pool of a protein of interest by rerouting it to the mitochondria (Robinson et al., 2010). AP-1 knocksideways produced a robust phenotype, with nearly 100 proteins lost twofold or more from CCV-enriched fractions, including hydrolase receptors (Hirst et al., 2012). We also carried out a GGA knocksideways and found that hydrolases and their receptors were lost over twofold from CCV-enriched fractions, while most other proteins were unaffected. Because GGAs and hydrolases were among the proteins that were lost from the CCV fraction in the AP-1 knocksideways, we proposed that AP-1 is bidirectional (Hirst et al., 2012), acting at the TGN together with GGAs to facilitate forward trafficking of hydrolase–receptor complexes and also acting at endosomes to retrieve empty receptors and other proteins back to the TGN (Fig. 1 A, Model 1).

Further support for a role for AP-1 in retrograde trafficking came from studies by Buser and Spiess using an elegant system they developed in which cells expressing GFP-tagged membrane proteins are incubated with derivatized anti-GFP nanobodies (Buser et al., 2018). Attaching tyrosine sulfation sequences to the nanobodies enabled trafficking from the plasma membrane to the TGN to be monitored because tyrosine sulfation occurs in the late Golgi. Nanobody sulfation in cells expressing GFP-tagged CDMPR or CIMPR was partially inhibited by AP-1 knocksideways, indicating a function in retrieval back to the Golgi (Buser et al., 2018). Surprisingly, however, knocking down AP-1 caused a ~2.5-fold increase in nanobody sulfation (Buser et al., 2022). The authors interpreted these findings as further evidence for a role for AP-1 in bidirectional trafficking and proposed that the increased nanobody sulfation in AP-1 knockdown cells was caused by increased residence time in the TGN, even though the endocytosed nanobody was found to accumulate in peripheral endosomes rather than in the Golgi region (Buser et al., 2022).

Studies in yeast and plant cells have provided further insights into AP-1 function, but also some confusing and/or contradictory findings. *Arabadopsis* has two genes encoding the AP-1 medium subunit, and knocking out the more strongly expressed gene caused defects in both the secretory and the vacuolar pathway, leading to the suggestion that AP-1 is required for trafficking to the plasma membrane (Park et al., 2013). However, subsequent studies showed that other proteins that are normally intracellular become mislocalized to the cell surface in AP-1-deficient *Arabidopsis* (Wang et al., 2014; Yan et al., 2021). In *S. cerevisiae*, studies dating back over 20 years show that AP-1 mutants mislocalize several late-Golgi resident proteins (Valdivia et al., 2002), and there is now a general consensus that the function of AP-1 in yeast is in retrieval back to the Golgi.

It seems unlikely that such an ancient and conserved piece of cellular machinery would have completely different functions in different eukaryotes, and we wondered whether some of the mutant phenotypes might be explained by indirect effects, even when they occurred very rapidly. For instance, our own observation that both hydrolases and GGAs are lost from HeLa cell CCV-enriched fractions in AP-1 knocksideways cells (Hirst et al., 2012) does not necessarily reflect a role for AP-1 in anterograde traffic. It could be that AP-1 is entirely retrograde but that it is needed to retrieve one or more molecules required by GGAs to facilitate forward transport (Fig. 1 B, Model 2). Thus, we wanted to look more directly at the AP-1 function. To this end, we have generated cells expressing tagged versions of both AP-1 and putative AP-1 cargo proteins. We have then used complementary approaches to investigate if, when, and where the cargo is getting packaged into AP-1 vesicles.

## Results

### Coexpression of tagged AP-1 and membrane proteins

We began by generating a cell line in which the AP-1 γ-adaptin gene, *AP1G1*, was tagged by inserting mRuby2 into its flexible hinge region (Fig. 2 A). Initially, we used gene editing; however, the mRuby2-positive clones all expressed considerably less γ-adaptin than wild-type cells (Fig. S1 A). This is most likely because HeLa cells are severely aneuploid with multiple copies of most genes, and only some of the *AP1G1* alleles had been tagged, while others had been disrupted by indels. Therefore, mRuby2-tagged γ-adaptin was introduced into the cells by retroviral transduction, and the endogenous gene was then deleted using CRISPR/Cas9. Clonal cell lines were confirmed by Western blotting to express only tagged γ-adaptin, at close to endogenous levels (Fig. 2 B). Because the cells became more heterogeneous with time, they were routinely sorted by flow cytometry before experiments were carried out (Fig. S1 B).

Additional constructs were then introduced into the cells by retroviral transduction. Cells were first stably transduced with a "Hook" construct, consisting of streptavidin with a KDEL sequence for ER retention (Fig. 2 A). Various candidate cargo constructs containing streptavidin-binding peptide (SBP) were then added so that their transit through the secretory pathway could be followed using the RUSH (Retention Using Selective Hooks) system (Boncompain et al., 2012). In the absence of biotin, these constructs remained in the ER, but the addition of biotin released them from the streptavidin-KDEL hook. Most of the SBP-containing proteins also had a GFP tag, allowing their

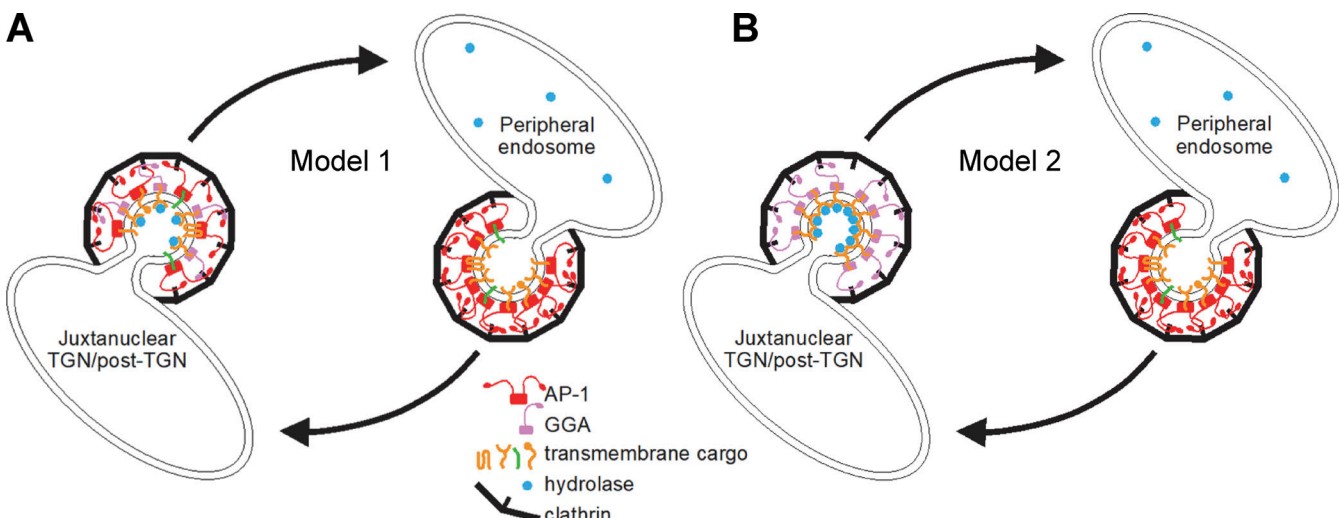

**Figure 1. Two models for AP-1 function. (A)** In Model 1, AP-1 acts together with GGAs at or near the TGN, packaging a number of proteins into clathrin-coated vesicles for trafficking to peripheral endosomes. These proteins include hydrolases and their receptors, which are sorted by GGAs, with AP-1 playing an auxiliary role. AP-1 has a further role in retrieving proteins from peripheral endosomes, including empty hydrolase receptors and TGN-resident membrane proteins such as syntaxins 6, 10, and 16. Adapted from Hirst et al. (2012). **(B)** In Model 2, AP-1 and GGAs act independently. GGAs sort hydrolases and their receptors at or near the TGN for trafficking to endosomes, while AP-1 acts exclusively at endosomes to retrieve proteins, including empty receptors, back to the TGN. One or more of these proteins may be needed by GGAs for forward transport.

appearance at the plasma membrane and subsequent endocytosis to be monitored with an anti-GFP nanobody (Fig. 2 A). Our previous work suggested that CDMPR is the most abundant AP-1 cargo protein (Hirst et al., 2012, 2015), so initially we used cells expressing SBP–GFP–CDMPR to set up the system.

To follow the fate of the CDMPR after exiting from the ER, the cells were incubated with biotin for various lengths of time over the course of 2 h, and in each case, HaloTag-conjugated nanobody and far-red HaloTag-binding dye were added for the final 30 min. The Western blots and fluorescent images in Fig. 2, B and C, show that by 15 min after biotin addition, most of the construct had reached the Golgi region, although it still had a similar electrophoretic mobility to the ER form. By 30 min, when most of the construct was still in the Golgi region, its mobility had partially shifted, indicative of terminal glycosylation. This shift was essentially complete by 40 min, and although the construct still localized mainly to the Golgi region, endocytosed nanobody could now be detected. By 90 min, the construct had reached steady state, with much of it found in more peripheral endosomes where it colocalized with the nanobody.

Several other Type I membrane proteins that had been identified as AP-1 cargo were then screened, and N-acetylglucosamine-1-phosphodiester alpha-N-acetylglucosaminidase (NAGPA, the M6P uncovering enzyme), which localizes to the Golgi region at steady state (Rorher and Kornfeld, 2001), was chosen for further studies. The lysosomal membrane protein LAMP1 was chosen as an example of a Type I membrane protein that does not appear to be cargo for AP-1, even though it has an AP-binding motif, YQTI, in its cytosolic tail (Peden et al., 2004). Like CDMPR, both of these proteins were tagged with SBP and GFP and transduced into cells already expressing Hook and tagged AP-1, using the same LTR promoter for moderate expression. Attempts to compare the expression of endogenous and tagged proteins

were hampered by the lack of suitable antibodies for Western blotting, but our results suggest that tagged LAMP1 is expressed more strongly than endogenous LAMP1 (data not shown), and the same is likely to be true for CDMPR and NAGPA. In addition, the accumulation of these proteins in the ER before the addition of biotin is likely to put a burden on the secretory pathway. Nevertheless, after 90 min in biotin, both constructs had reached their expected destinations, indicating that the cell is able to sort the proteins correctly (Fig. 3 A). NAGPA was tightly juxtanuclear, presumably localizing to the TGN (Rohrer and Kornfeld, 2001), while LAMP1 was more punctate, with larger spots than either CDMPR or NAGPA, presumably localizing to late endosomes and lysosomes (Peden et al., 2004). Cells expressing both constructs had internalized a substantial amount of anti-GFP nanobody by 90 min (Fig. 3 A).

**Comparative localization of AP-1 and membrane proteins**
As a first test for AP-1-mediated trafficking of the three constructs, we transiently transfected cells with gadkin, a binding partner for the AP-1 γ appendage domain (Neubrand et al., 2005). Overexpression of gadkin stabilizes the association of AP-1 with membranes and causes AP-1 and associated proteins to accumulate at the cell periphery (Schmidt et al., 2009). Previous studies have shown that endogenous CDMPR is affected by gadkin overexpression, while there is little or no effect on endogenous LAMP1 (Schmidt et al., 2009). Similarly, Fig. 3 B shows that after 90 min in biotin, SBP-GFP-CDMPR colocalized with gadkin and AP-1 at peripheral foci (arrowheads), while SBP-GFP-LAMP1, for the most part, did not. SBP-GFP-NAGPA also showed strong colocalization with gadkin. Thus, these three constructs were used as model proteins to investigate the function of AP-1. Two of the constructs, SBP-GFP-CDMPR and SBP-GFP-NAGPA, appear to be AP-1-dependent, but they have

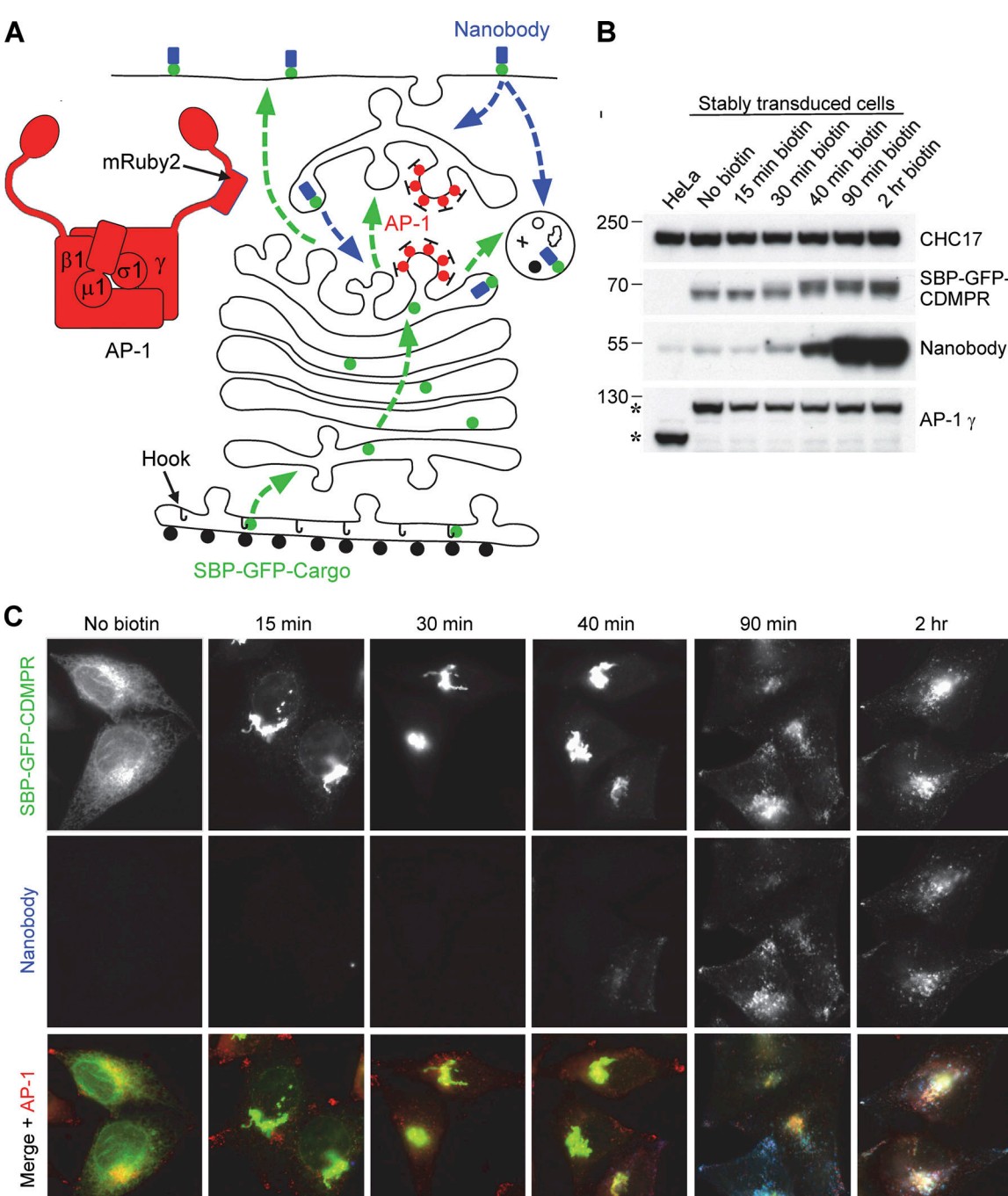

**Figure 2. System for investigating outgoing and incoming membrane traffic. (A)** Schematic diagram of the system. A HeLa cell line was generated in which the endogenous AP-1 γ-adaptin subunit, encoded by *AP1G1*, was replaced by a mRuby2-tagged version. The cells were then stably transduced with a "Hook" construct, consisting of streptavidin with a KDEL sequence for ER retention, and one or more candidate cargo constructs, containing streptavidin-binding peptide (SBP) to keep the constructs in the ER until the addition of biotin. Most of these constructs were also tagged with GFP so that their appearance at the plasma membrane and subsequent endocytosis could be monitored with an anti-GFP nanobody. The green dotted lines with arrows show the pathways that might be taken by the SBP-GFP-cargo proteins after the addition of biotin, and the blue dotted lines with arrows show the pathways that might be taken by the nanobody after it has bound to SBP-GFP-cargo proteins on the cell surface. **(B)** Western blot of cells shown schematically in A, in which the SBP- and GFP-tagged cargo protein was the cation-dependent mannose 6-phosphate receptor (CDMPR). Addition of biotin causes a shift in the mobility of SBP-GFP-CDMPR (visualized with anti-GFP) as it moves through the Golgi apparatus. HaloTag-conjugated nanobody was added to all the cells for the final 30 min and it started to show a clear increase above background levels (which is most likely due to fluid-phase uptake) after 40 min in biotin. The AP-1 γ subunit is expressed at similar levels in the wild-type and transduced cells, but it has different mobilities (asterisks) because of the presence or absence of the mRuby2 tag. Clathrin heavy chain (CHC17) was used as a loading control. **(C)** Fluorescent images of the cells shown in B. After 15 min in biotin, most of the SBP-GFP-CDMPR has moved from the ER to the Golgi region, and it stays mostly Golgi-localized until after 40 min. Endocytosed nanobody starts to be detectable at 40 min. Scale bar: 10 µm. Source data are available for this figure: SourceData F2.

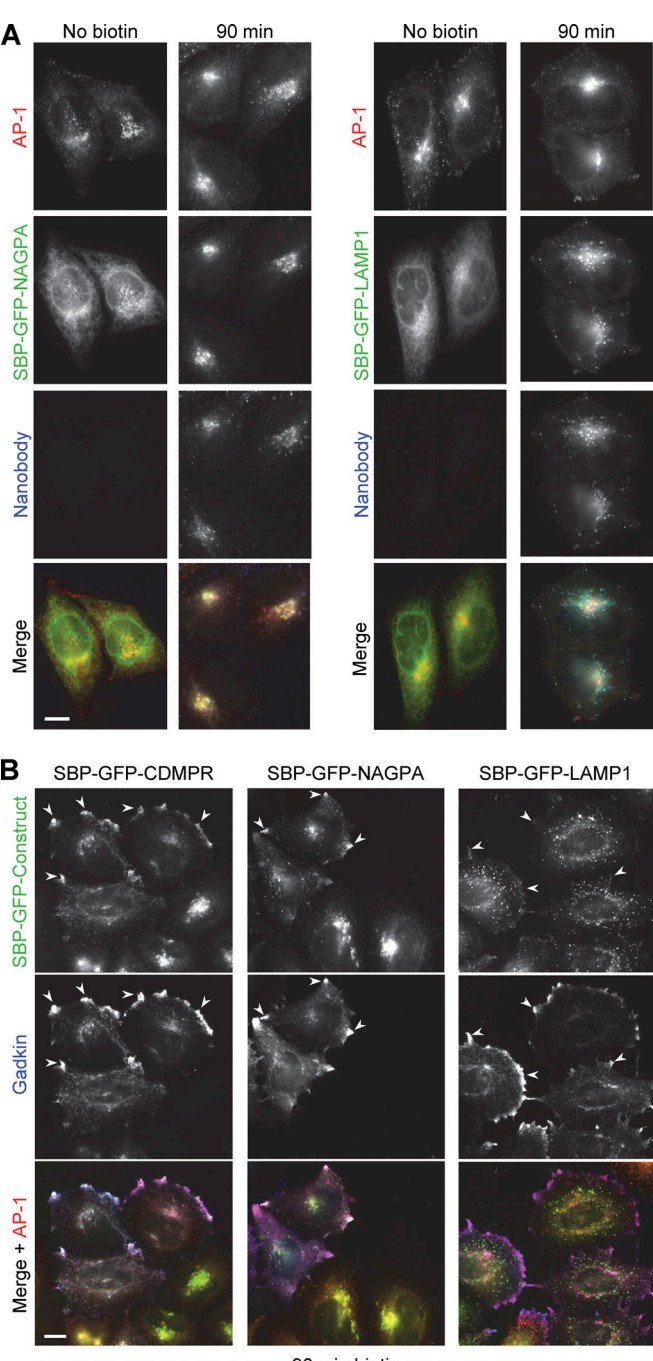

Figure 3. **Comparison of SBP-GFP-tagged CDMPR, NAGPA, and LAMP1.** **(A)** Like SBP-GFP-CDMPR, both SBP-GFP-NAGPA and SBP-GFP-LAMP1 are retained in the ER in the absence of biotin, and there is no detectable uptake of nanobody. After 90 min in biotin, both have reached their normal steady state localization and have endocytosed substantial amounts of nanobody, which was added for the final 30 min. Scale bar: 10 μm. **(B)** Cells expressing SBP-GFP-CDMPR, SBP-GFP-NAGPA, or SBP-GFP-LAMP1 were transfected with HaloTag-gadkin and then treated with biotin for 90 min. Both SBP-GFP-CDMPR and SBP-GFP-NAGPA show strong colocalization with gadkin at the cell periphery, while SBP-GFP-LAMP1 does not. This colocalization with gadkin is indicative of an association with AP-1. Scale bar: 10 μm.

distinct steady-state distributions, while the third construct, SBP-GFP-LAMP1, appears to be AP-1-independent.

To determine whether AP-1 is associated with any of these constructs when they exit the Golgi apparatus, we treated cells

with biotin for 20–35 min and then imaged the cells using Lattice SIM super-resolution microscopy. As has been previously reported (Chen et al., 2017), CDMPR left the Golgi mainly in vesicular carriers, and we saw no obvious colocalization of these carriers with AP-1 (data not shown). In contrast, NAGPA was found mainly in tubular carriers, typical of proteins that are transported from the TGN to the plasma membrane, such as LAMP1 (Chen et al., 2017). Indeed, when SBP-GFP-LAMP1 and SBP-HaloTag-NAGPA were coexpressed, they could be seen to exit the Golgi in the same tubules (Fig. 4 A and Video 1). Although it is possible that overexpression of NAGPA caused more of it to leak LAMP1-containing tubules, it is known that many TGN-resident proteins, including NAGPA, can leave the Golgi and then return (Chapman and Munro, 1994; Rohrer and Kornfeld, 2001), and the steady state localization of the tagged NAGPA was still mainly in the Golgi region (Fig. 3 A).

Most unexpectedly, the NAGPA-containing tubules leaving the Golgi were frequently decorated with AP-1 puncta (Fig. 4, B and C; and Videos 2 and 3). Unlike NAGPA and LAMP1, CDMPR was not concentrated in the tubules, but small amounts could sometimes be seen (Fig. 4 C and Video 3). To rule out the possibility that AP-1 was recruited onto these tubules because they contained excess amounts of NAGPA, we also imaged cells expressing just SBP-GFP-LAMP1 and again saw AP-1 puncta on LAMP1-containing tubules leaving the Golgi (Fig. S1 C and Video 4).

AP-1 puncta were found not only in the Golgi region but also at the cell periphery. To find out whether these puncta corresponded to endosomes, we loaded cells expressing mRuby2-tagged AP-1 and SBP-GFP-NAGPA with Alexa Fluor 647-conjugated transferrin and then treated the cells with biotin for 20–35 min. Live cell imaging showed that many or even most of the peripheral AP-1 puncta were associated with transferrin-containing spots and tubules (Fig. 4 D; and Videos 5 and 6).

All of these structures were extremely dynamic. AP-1 has been shown to be removed from vesicles within seconds after scission (Kural et al., 2012), so it was not possible to determine whether any of the NAGPA, CDMPR, or transferrin was getting incorporated into AP-1 vesicles, especially because the number of copies of each cargo protein in an individual vesicle would be difficult to discern above background. Therefore, we developed a complementary assay, which would enable us to get a snapshot of these very transient transport intermediates, without interference from the rest of the cell.

## Single-vesicle analysis

Our assay, which we call single-vesicle analysis (SVA), is based on widefield microscopy of isolated CCVs, and is shown schematically in Fig. 5 A. CCV-enriched fractions were prepared from cells co-expressing mRuby2-tagged AP-1, Hook, and SBP-GFP-tagged membrane proteins, which had been incubated with biotin for 0, 30, 40, or 120 min, and with HaloTag-nanobody for the final 30 min. Varying dilutions of the CCV fraction were aliquoted onto slides that had been precoated with 100-nm fluorescent beads, to act as a fiducial marker for channel alignment. The beads are of similar size and fluorescence intensity to CCVs, but they fluoresce in four channels instead of three, so they could be eliminated from our final analysis. An

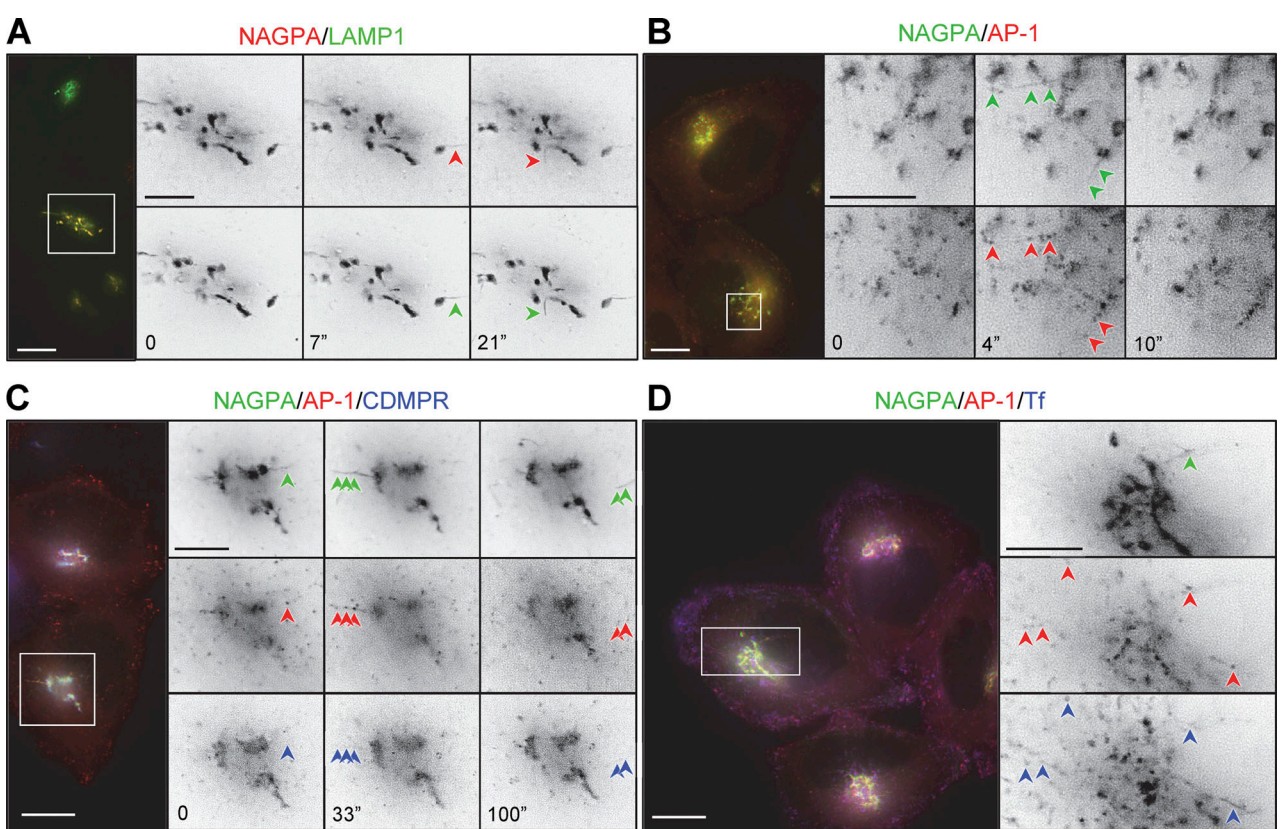

Figure 4. **Live imaging of cells coexpressing mRuby2-tagged AP-1 and one or more tagged membrane proteins. (A)** Frames from Video 1, showing cells co-expressing SBP-HaloTag-NAGPA (red) and SBP-GFP-LAMP1 (green), treated with biotin for 20 min. LAMP1 and NAGPA can be seen leaving the Golgi in the same tubules (arrowheads). The number of seconds between frames is indicated. Scale bar: 10 μm. **(B)** Frames from Video 2, showing cells coexpressing mRuby2-AP-1 (red) and SBP-GFP-NAGPA (green), treated with biotin for 33.5 min. NAGPA-positive tubules are decorated with AP-1 puncta. The number of seconds between frames is indicated. Scale bar: 10 μm. **(C)** Frames from Video 3 show cells coexpressing mRuby2-AP-1 (red), SBP-HaloTag-NAGPA (green), and SBP-GFP-CDMPR (blue), treated with biotin for 27 min. Although most of the CDMPR leaves the Golgi in vesicles rather than tubules, small amounts of CDMPR can sometimes be seen in NAGPA-positive tubules. These tubules are frequently decorated with AP-1 puncta (arrowheads). The number of seconds between frames is indicated. Scale bar: 10 μm. **(D)** A frame from Video 5, showing cells coexpressing mRuby2-AP-1 (red) and SBP-GFP-NAGPA (green), fed far-red transferrin (blue) for 1 h, and treated with biotin for 27 min. AP-1 puncta can be seen on both NAGPA-positive tubules (upper right arrowhead) and transferrin-positive tubules (other arrowheads). Transferrin-containing tubules at the cell periphery are frequently decorated with AP-1 puncta (see also Video 6). Scale bars: 10 μm for color images, 5 μm for black and white images.

example of such an experiment is shown in Fig. 5 B and in Fig. S2. At least ten images were collected for every time point, and each experiment was repeated three times (Fig. S3). To analyze the dataset, we developed a Python-based pipeline, which first uses the beads for homographic channel alignment, then discerns each punctate structure through the application of the Laplacian of Gaussian filter and quantifies the intensity of each fluorophore. The beads are then excluded from subsequent analyses. Thresholds were established by using single-channel controls, facilitating the binarization of data to determine the presence or absence of each fluorophore. This methodology allowed every vesicle to be categorized based on the presence of AP-1 and/or cargo.

Fig. 5 C shows a Western blot (equal protein loading) of homogenates and CCV-enriched fractions from cells expressing SBP-GFP-CDMPR. In the lanes containing CCV fractions, there is strong enrichment of both construct and nanobody after 30 min in biotin, which continues to increase up to 2 h. This is consistent with packaging into AP-1 vesicles, but the Western blot only shows that they are enriched in the CCV-containing pellet,

which contains other small structures as well. Using single-vesicle analysis (Fig. 5 D and Fig. S3 A), we were able to quantify the percentage of AP-1-positive vesicles that also contained cargo and/or nanobody. By 2 h, when the CDMPR had reached steady state, about three-quarters of the AP-1 vesicles were found to contain CDMPR, and nearly half of these also contained the endocytosed nanobody. There were essentially no nanobody-containing spots that were not also positive for CDMPR, as expected because the nanobody gets into the cell by piggybacking onto the tagged CDMPR. These percentages are probably an underestimate, as the expression of both AP-1 and cargo constructs was somewhat heterogeneous (see Fig. S1 B), so there may have been additional AP-1-positive spots containing tagged CDMPR that were below the limit of detection. In addition, our widefield imaging system appears to be relatively insensitive to far-red (the signal from the fluorescent beads was weaker in the far-red channel than in the other three channels), so the percentages of CDMPR puncta that contained nanobody may also have been underestimated.

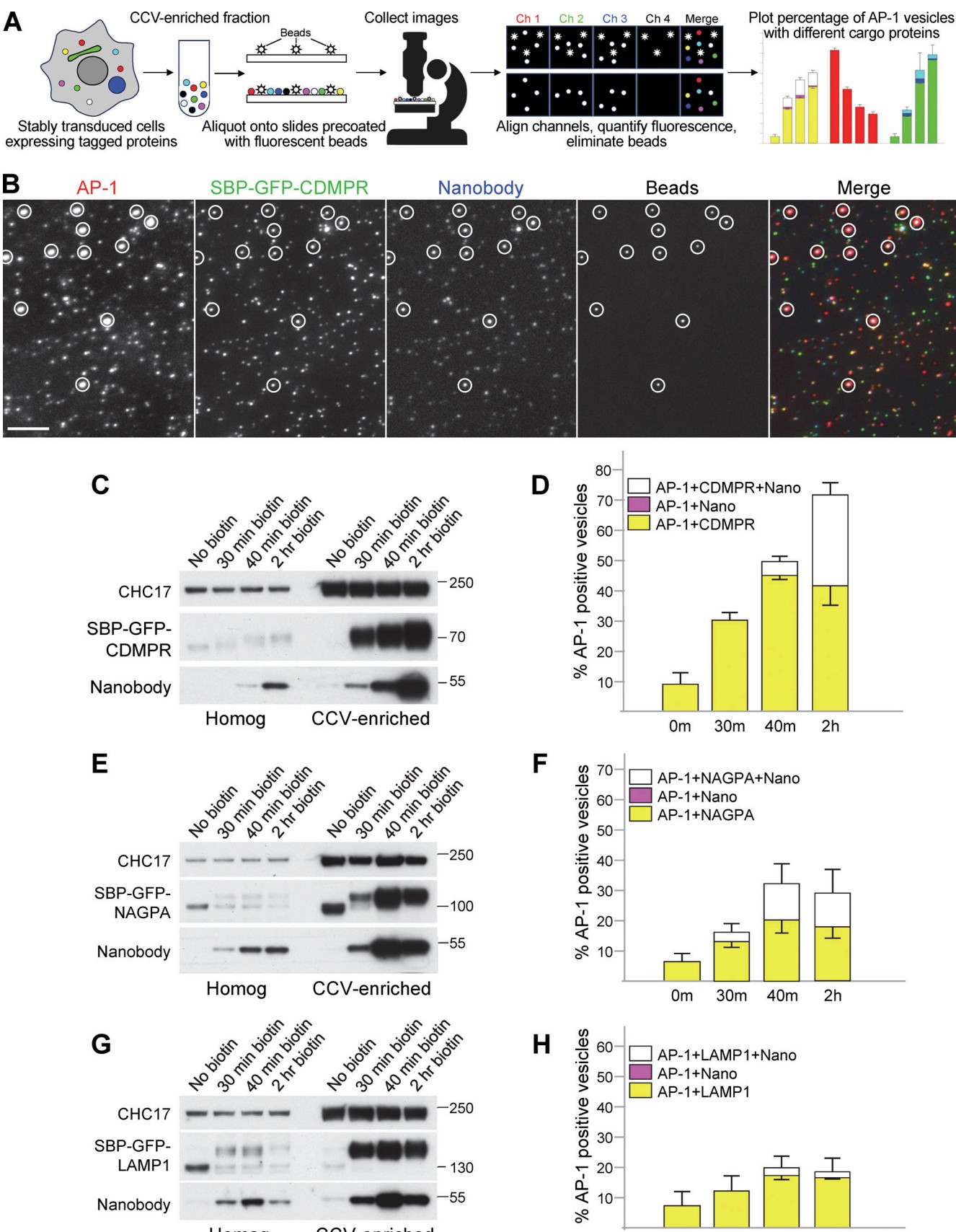

Figure 5. **Single vesicle analysis of cells coexpressing mRuby2-tagged AP-1 and GFP-tagged membrane proteins. (A)** Overview of the technique. A CCV-enriched fraction is prepared from stably transduced cells and aliquoted onto slides that were precoated with 100-nm beads. The beads fluoresce in four

wavelengths and are used as fiducial markers for channel alignment. Widefield images are collected in four channels and analyzed using a newly written script. The analysis includes channel alignment, quantification of fluorescence in each spot, and elimination of the beads, which are the only particles that fluoresce in Channel 4. Data are plotted as the percentage of AP-1-containing spots (Channel 1) that also contain another protein or proteins. **(B)** A CCV-enriched fraction was prepared from cells co-expressing mRuby2-tagged AP-1 and SBP-GFP-tagged CDMPR, treated with biotin for 2 h, and incubated with HaloTag-conjugated nanobody for the final 30 min. Aliquots were spotted onto slides that had been precoated with beads. The figure shows cropped images in each wavelength, with the beads circled. The full image can be seen in Fig. S2. AP-1 fluoresces in red, CDMPR in green, and nanobody in far-red, shown as blue in the merged image. Scale bar: 5 µm. **(C)** Western blot from an experiment similar to the one in B, showing the whole cell homogenate and CCV-enriched fractions for four different time points in biotin. CHC17 was used as a loading control. **(D)** Single vesicle analysis of the experiment is shown in C. The data represent the means from three independent pooled experiments, with at least 10 images analyzed in each experiment for each condition, and 1,000–10,000 discrete spots per image. The error bars indicate the standard deviation. The presence of CDMPR in AP-1 vesicles continues to increase throughout the time course, so that by 2 h in biotin, about three-quarters of the AP-1 vesicles contain detectable CDMPR, and nearly half of these also contain detectable endocytosed nanobody. There are no vesicles containing nanobodies that do not also contain CDMPR, as expected because the nanobody enters the cell by piggybacking on the CDMPR. Complete datasets for all the single vesicle analysis experiments are shown in Fig. S3. **(E)** Western blot of cell homogenates and CCV-enriched fractions from cells expressing SBP-GFP-NAGPA, treated with biotin for varying lengths of time and incubated with nanobody for the final 30 min. CHC17 was used as a loading control. **(F)** Single-vesicle analysis of the experiment is shown in E. The data represent the means from three independent pooled experiments, with at least 10 images analyzed in each experiment for each condition, and 1,000–10,000 discrete spots per image. The error bars indicate the standard deviation. The presence of NAGPA in AP-1 vesicles plateaus at 40 min, with about one-third of the AP-1 vesicles containing detectable NAGPA. Nearly half of these also contain detectable endocytosed nanobody, which accumulates more quickly in NAGPA-expressing cells than in CDMPR-expressing cells, consistent with NAGPA leaving the Golgi in LAMP1-positive tubules (see Fig. 4 A). **(G)** Western blot of cell homogenates and CCV-enriched fractions from cells expressing SBP-GFP-LAMP1, treated with biotin for varying lengths of time and incubated with nanobody for the final 30 min. CHC17 was used as a loading control. **(H)** Single-vesicle analysis of the experiment shown in G. The data represent the means from three independent pooled experiments, with at least 10 images analyzed in each experiment for each condition, and 1,000–10,000 discrete spots per image. The error bars indicate the standard deviation. Relatively little LAMP1 accumulates in AP-1 vesicles. Source data are available for this figure: SourceData F5.

NAGPA showed somewhat different behavior (Fig. 5, E and F; and Fig. S3 B). First, while CDMPR levels in AP-1 vesicles continued to rise from 40 min to 2 h, NAGPA reached a plateau at 40 min. Second, after 2 h in biotin, most AP-1 vesicles contained detectable CDMPR, whereas only ∼30% had detectable NAGPA. Third, there was a higher association of endocytosed nanobody with AP-1/NAGPA-positive vesicles than with AP-1/CDMPR-positive vesicles after 30 and 40 min in biotin. However, nanobody uptake increased dramatically between 40 min and 2 h in cells expressing SBP-GFP-CDMPR, whereas no such increase occurred in cells expressing SBP-GFP-NAGPA. These results, together with our imaging data, suggest that at least some of the CDMPR and NAGPA get packaged into AP-1 vesicles shortly after leaving the Golgi. However, our results also suggest that some of the NAGPA is transported to the plasma membrane via the constitutive secretory pathway, where it is then endocytosed and subsequently packaged into AP-1 vesicles. Our nanobody uptake experiments indicate that CDMPR gets to the plasma membrane more slowly, with very little nanobody accumulated after 40 min in biotin, but by the time the CDMPR has reached steady state, much of it has picked up nanobody.

LAMP1 did not show much enrichment in AP-1 vesicles (Fig. 5, G and H; and Fig. S3 C), even though it was present in the same AP-1-studded tubules as NAGPA (Fig. 4, A and C). After 40 min in biotin, SBP-GFP-LAMP1 was enriched in the CCV-containing pellet, but we never found more than ∼20% of AP-1 vesicles showing colocalization with LAMP1. Some of this colocalization may be non-specific because for all our constructs, we saw close to 10% colocalization with AP-1 even in the absence of biotin. It is also possible that overexpression caused more LAMP1 to spill into AP-1 vesicles.

### Heterogeneity of AP-1 vesicles

Our SVA results show that both CDMPR and NAGPA are packaged into AP-1 vesicles, but are these the same vesicles, or are there different types of vesicles for different types of cargo? To address this question, we generated cells coexpressing SBP-GFP-CDMPR and SBP-HaloTag-NAGPA, as well as Hook and tagged AP-1. Despite having distinct steady-state distributions and leaving the Golgi in different carriers, ∼80% of AP-1 vesicles carrying NAGPA also contained CDMPR (as shown in Fig. 6, B and C, and in Fig. S3 D). The percentage of AP-1 vesicles that were positive for NAGPA was less with HaloTag than with the GFP tag. This is probably due not only to the relative insensitivity of our system to far-red but also to the inherently lower expression of HaloTag constructs (Fig. S4 A). Thus, we suspect that there are additional HaloTag-NAGPA-containing vesicles that were below the limit of detection. To test whether the vesicles that contained detectable NAGPA but not CDMPR might be due solely to differences in expression levels, we transduced cells with both SBP-GFP-CDMPR and SBP-HaloTag-CDMPR. In these cells, there was essentially complete overlap between the GFP and HaloTag signals (Fig. S4 B). However, in isolated vesicles, there were not only AP-1/GFP-positive spots without detectable HaloTag but also AP-1/HaloTag-positive spots without detectable GFP (Fig. 6 D and Fig. S3 E). These accounted for ∼10% of the AP-1/HaloTag-positive spots, and we assume that this is due to cell heterogeneity. Thus, a very small fraction of the AP-1 vesicles contains NAGPA but not CDMPR, and a larger fraction contains CDMPR but not NAGPA. We think that this larger fraction of vesicles containing CDMPR but not NAGPA is a reflection of the more widespread distribution of CDMPR, mirroring the distribution of AP-1, which is found at multiple locations in the cell, whereas most of the NAGPA is in the juxtanuclear region.

There are many reports, including the present study, that some AP-1 is associated with tubular endosomes containing internalized transferrin (Futter et al., 1998; Peden et al., 2004; Fig. 4 D; and Videos 6 and 7). To look for possible overlap between internalized transferrin and newly synthesized CDMPR,

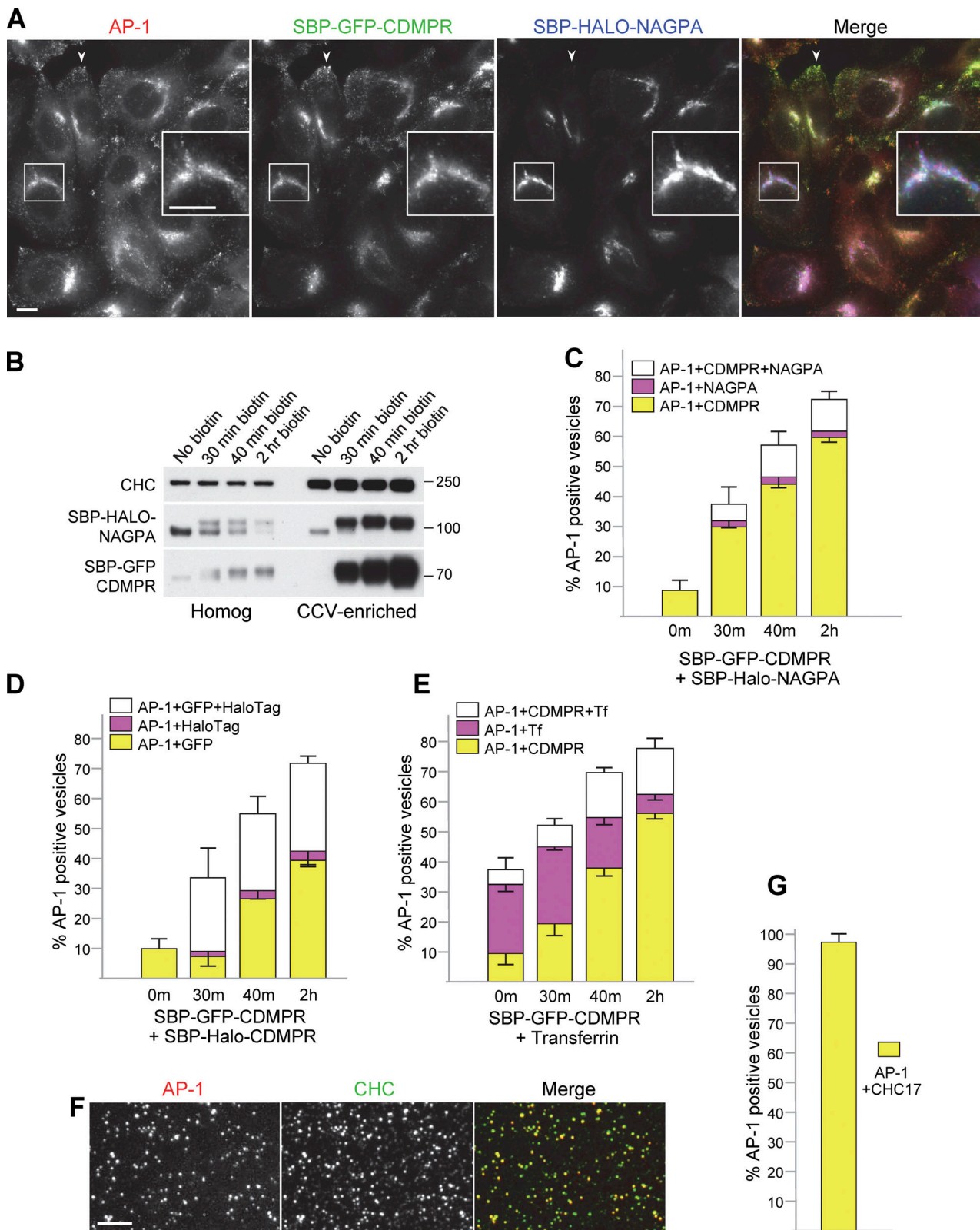

Figure 6. **Heterogeneity of AP-1 vesicles. (A)** Fluorescent images of cells coexpressing SBP-GFP-CDMPR and SBP-HaloTag-NAGPA after 90 min in biotin. Although both localize to juxtanuclear membranes, the fine details are different (inset), and only CDMPR localizes to peripheral membranes, where AP-1 is also found (arrowheads). Scale bars: 10 μm. **(B)** Western blot of cells co-expressing SBP-GFP-CDMPR and SBP-HaloTag-NAGPA, showing the whole cell homogenate and CCV-enriched fractions for four different time points in biotin. CHC17 was used as a loading control. **(C)** Single-vesicle analysis of the experiment shown in B. The data represent the means from three independent pooled experiments, with at least 10 images analyzed in each experiment for each condition, and 1,000–10,000 discrete spots per image. The error bars indicate the standard deviation. Most of the NAGPA-containing AP-1 vesicles (∼80%) also contain

CDMPR, but not vice versa. The percentages are deduced from the white and magenta portions of the bars: in every case, the white portion is ∼4× taller than the magenta portion. **(D)** Single vesicle analysis of cells co-expressing SBP-GFP-CDMPR and SBP-HaloTag-CDMPR, showing the means from three independent pooled experiments, with at least 10 images analyzed in each experiment for each condition, and 1,000–10,000 discrete spots per image. The error bars indicate the standard deviation. The differences are most likely due to differences in expression levels because the localization patterns of SBP-GFP-CDMPR and SBP-HaloTag-CDMPR are virtually identical (see Fig. S4 B). **(E)** Single vesicle analysis of cells expressing SBP-GFP-CDMPR, treated with biotin for various lengths of time, and in every case fed far-red transferrin for 1 h and then processed immediately. The data represent the means from three independent pooled experiments, with at least 10 images analyzed in each experiment for each condition, and 1,000–10,000 discrete spots per image. The error bars indicate the standard deviation. The percentage of AP-1 vesicles that contain both CDMPR and transferrin increases with time in biotin. **(F)** Fluorescent image of the CCV-enriched fraction from cells expressing mRuby2-tagged AP-1 and labeled after fixation with an antibody against CHC17. Scale bar: 5 µm. **(G)** Single-vesicle analysis of experiments similar to the one shown in F, with at least 10 images analyzed in each of three experiments, and 1,000–10,000 discrete spots per image. The error bars indicate the standard deviation. Nearly all of the AP-1 vesicles are also positive for CHC17. Source data are available for this figure: SourceData F6.

we carried out a biotin time course on cells expressing SBP-GFP-CDMPR, while at the same time incubating them with far-red transferrin for 1 h. About 30% of the AP-1 vesicles were positive for endocytosed transferrin, and the percentage of these vesicles that also contained CDMPR increased with time from ∼20% at 30 min to ∼70% at 2 h (Fig. 6 E and Fig. S3 F). This indicates that there is more CDMPR in transferrin-containing endosomes when the CDMPR has reached steady state than when it first leaves the Golgi apparatus. Nevertheless, the overlap was always less than the overlap between CDMPR and NAGPA. Together, these data show that AP-1 vesicles are a heterogeneous population, presumably reflecting the widespread distribution of AP-1.

Are some of the AP-1 vesicles in our fraction also coated with clathrin? We found that we could fix the vesicles after aliquoting them onto slides and then look for the presence of other proteins by immunofluorescence. Nearly 100% of the AP-1 vesicles were positive for clathrin, although not vice versa (Fig. 6, F and G), as expected because clathrin is present on other vesicles as well, such as those that bud from the plasma membrane.

### Comparison of AP-1 and GGA2

Membrane proteins like CDMPR and NAGPA can travel in both the anterograde and the retrograde direction, so the results presented so far do not answer the question of AP-1 directionality. Two potential markers for the forward pathway are lysosomal hydrolases and GGAs (see Fig. 1). Unfortunately, when we tried to tag several different hydrolases with GFP, both with and without SBP, we found that all of the constructs were unable to leave the ER.

Our studies on GFP-tagged GGA2 were more successful. As previously reported (Boman et al., 2000; Dell'Angelica et al., 2000; Hirst et al., 2000), both AP-1 and GGA2 were concentrated in the juxtanuclear area and there were also puncta in more peripheral parts of the cell, which were more numerous for AP-1. However, there was relatively little apparent colocalization between the AP-1 and GGA2 puncta (Fig. 7). Moreover, by live cell imaging using simultaneous capture, the AP-1 and GGA2 puncta could be seen to be mainly on separate trajectories (Video 7). These observations are inconsistent with Model 1 (Fig. 1 A), which proposes that AP-1 and GGAs cooperate in the same budding vesicle. They are also inconsistent with an earlier model, which proposed that GGAs and AP-1 act sequentially (Doray et al., 2002), because we did not see any clear examples of an individual spot changing from GGA2-positive to AP-1-positive.

Although GGA2-positive puncta were easily detectable in intact cells, when a CCV-enriched fraction was prepared from such cells, the GFP signal was below the limit of detection. This

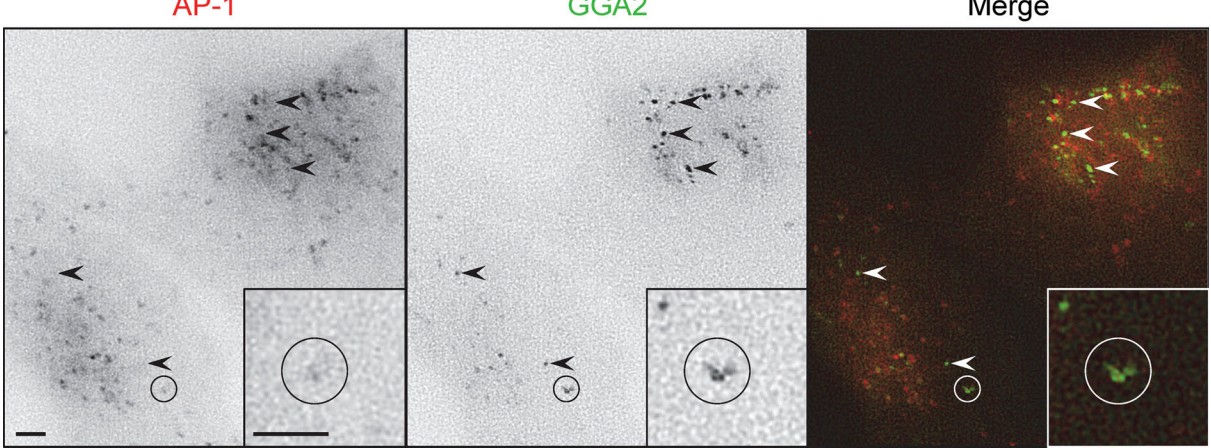

Figure 7. **Stills from Video 7 showing cells co-expressing mRuby2-tagged AP-1 and GFP-tagged GGA2.** Arrowheads indicate some of the puncta that are positive for GGA2 only. The circled area indicates a moving structure that is positive for both AP-1 and GGA2, but the two localize to different regions. Scale bars: 2 µm.

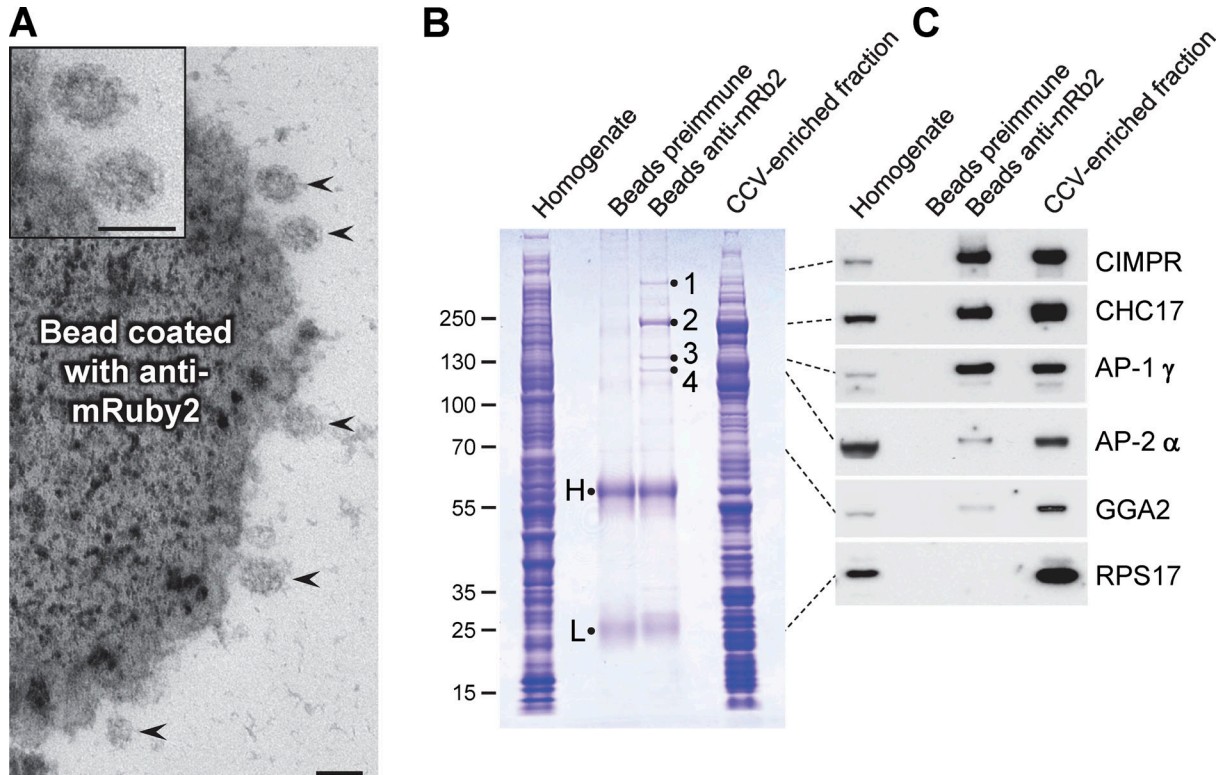

Figure 8. **Capture of AP-1 vesicles using magnetic beads. (A)** A CCV-enriched fraction was prepared from HeLa cells expressing mRuby2-tagged AP-1 γ, and then the AP-1-positive vesicles were captured just before the final centrifugation step using a rabbit antiserum against mRuby2 followed by protein A-coated magnetic beads. CCVs (arrowheads) can be seen to be associated with the beads; the two top ones are shown magnified in the inset. Scale bars: 100 nm. **(B)** Coomassie blue-stained gel of the whole cell homogenate, proteins captured on magnetic beads, and the CCV-enriched fraction from cells expressing mRuby2-tagged AP-1 γ. As a control for specificity, the preimmune serum from the same rabbit was also incubated with protein A beads. The samples containing beads and the CCV-enriched fraction were made up to the same volume so the efficiency of capture could be assessed, and the sample containing cell homogenate was made up to the same protein concentration as the sample containing the CCV fraction. H and L refer to immunoglobulin heavy and light chains; the bands labeled 1–4 were excised and found to correspond to CIMPR, CHC17, the AP-1 γ subunit, and the AP-1 β1 subunit, respectively. **(C)** Western blot of the samples shown in B probed with various antibodies. The dotted lines indicate the position on the gel for each antigen. In addition to the expected proteins, a small amount of AP-2 was also captured by the beads. GGA2 has a similar profile to AP-2, with relatively little captured by the beads. Source data are available for this figure: SourceData F8.

is most likely because the association of GGAs with membranes is relatively labile (Hirst et al., 2001). Therefore, we developed a complementary approach to analyze the protein composition of AP-1 vesicles.

**Vesicle isolation by antibody capture**

Using a new rabbit antiserum against mRuby2, together with protein A–coated magnetic beads, we devised a method for capturing AP-1 vesicles. First, we carried out our usual CCV fractionation protocol, but then before the final centrifugation step, we added beads that had been preincubated with either anti-mRuby2 antiserum or preimmune serum from the same rabbit. The beads were then either prepared for electron microscopy or treated with SDS-containing sample buffer to elute bound proteins.

Fig. 8 A shows an electron micrograph of the anti-mRuby2 beads. The beads can be seen to be covered with ~100 nm vesicles (arrowheads), and in most cases, a clathrin coat can be discerned. By SDS PAGE (Fig. 8 B), there are several high molecular weight bands in the lane containing the anti-mRuby2

beads that are not visible in the lane containing preimmune beads. The bands numbered 1–4 were excised and analyzed by mass spectrometry, and they were found to correspond respectively to CIMPR, clathrin heavy chain (CHC17), the AP-1 γ subunit, and the AP-1 β1 subunit.

Western blots of the samples (Fig. 8 C) confirmed and extended these observations. Both AP-1 γ and CIMPR were captured very efficiently, with similar band intensities in the bead-captured lane and the CCV-enriched fraction lane, indicating that most of the protein had bound to the beads. Clathrin heavy chain was captured less efficiently, even though the Coomassie blue-stained gel indicates that it is the most abundant protein pulled out by the beads. This is consistent with our immunofluorescence results (Fig. 6 G), which show that much of the clathrin in the CCV-enriched fraction is not associated with AP-1. There was much less GGA2 captured by the beads, and in fact, its profile was very similar to that of AP-2, which we did not expect to see at all, because it is associated with the population of CCVs that bud from the plasma membrane rather than intracellular membranes. The ribosomal protein RPS17, which is very

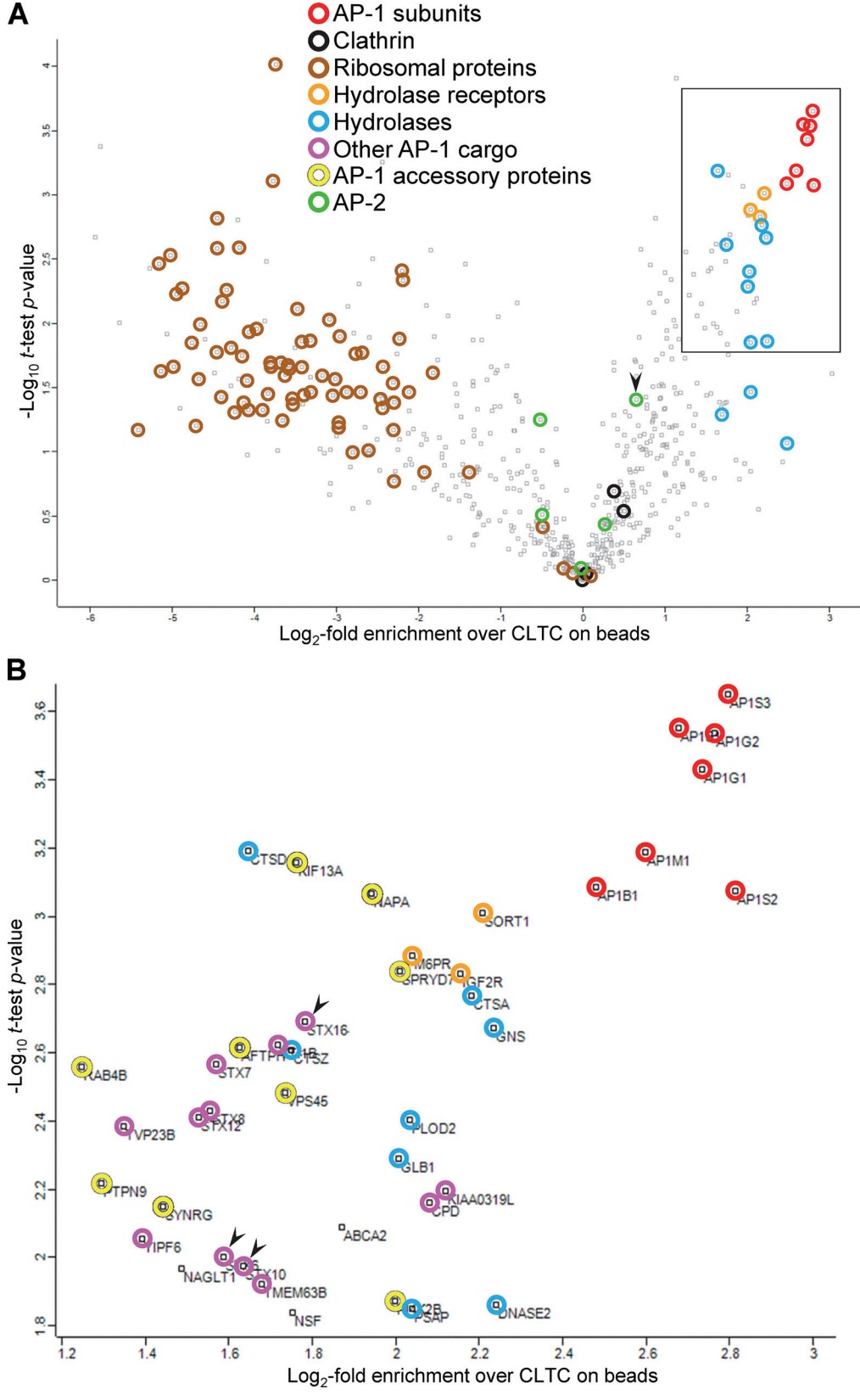

**Figure 9. Proteomic analysis of the bead-captured samples.** CCV-enriched fractions were prepared from HeLa cells expressing mRuby2-tagged AP-1 γ, grown in either SILAC heavy (H) or SILAC light (L) medium. AP-1-positive vesicles were captured from the H fraction before the final centrifugation step, while the L fraction was centrifuged to provide an input reference. Heavy-to-light ratios were calculated as a measure of enrichment in the bead-captured samples over the input and statistical analysis was performed for 632 proteins that passed quality control. **(A)** Data are presented as a volcano plot. For each protein, a one-sample *t* test was performed against a hypothetical ratio defined as the mean ratio for CLTC, the most abundant protein in both the H and L samples (see

Fig. S5). The proteins on the right side of the plot are potential components of AP-1 vesicles, with the most enriched hits being AP-1 subunits. The three hydrolase receptors and various hydrolases also score highly. Proteins on the left side are unlikely to be components of AP-1 vesicles, and they include ribosomal proteins. AP-2 subunits are generally to the left of clathrin, with the exception of AP2B1 (arrowhead), which is known to associate promiscuously with AP-1 (Page and Robinson, 1995). Other AP-1 cargo and accessory proteins are not indicated in this plot. **(B)** Protein identities in the boxed region in A, which contains the most enriched hits. The proteins shown in color were also hits in our previous knocksideways analyses (Hirst et al., 2012, 2015). The arrowheads indicate the syntaxins STX6, STX10, and STX16. The complete dataset is available in Table S1. Table S2 contains proteomic data from unlabeled cells, in which the CCV-enriched fraction was incubated with beads coated with either anti-mRuby or preimmune serum, as further confirmation of the specificity of the AP-1 vesicle capture.

abundant in the CCV-enriched fraction, was generally undetectable in the bead samples.

## AP-1 vesicle composition

For a more comprehensive and quantitative analysis of the protein content of AP-1 vesicles, we performed mass spectrometry on SILAC-labeled cells. Preparations were carried out in parallel on cells incubated with either heavy or light amino acids, but before the final step, AP-1 vesicles were captured from the heavy preparation with anti-mRuby2 beads, while the light preparation was centrifuged one last time to provide a reference CCV-enriched fraction. The heavy-to-light ratio for each protein is a measure of how efficiently that protein was captured by the beads. For instance, the mean ratio for AP-1 γ was 0.81 (i.e., >80% of the input), confirming the high efficiency of our AP-1 vesicle capture. The experiment was carried out four times, and the mean heavy-to-light ratio of CLTC (CHC17) was used as a reference. Our rationale was that CLTC is the most abundant protein in both the CCV-enriched input sample and the bead-captured sample (Fig. S5), but not all of the clathrin in the input sample is expected to associate with AP-1; thus, any protein with a heavy-to-light ratio greater than that of clathrin heavy chain is a potential component of AP-1 vesicles. The results are presented as a volcano plot in Fig. 9, with the enrichment of each protein relative to CLTC on the x-axis and the statistical significance on the y-axis. The AP-1 complex itself is the top hit, and the three hydrolase receptors also score very highly. Lysosomal hydrolases are also major hits. Other strong hits, shown in Fig. 9 B, include a number of TGN resident proteins, such as carboxypeptidase D, KIAA0319L, and the late Golgi SNAREs syntaxins 6, 10, and 16 (arrowheads), as well as peripheral membrane proteins such as KIF13A, VPS45, and SPRYD7, all of which also scored highly in our AP-1 knocksideways dataset (Hirst et al., 2012). In contrast, ribosomal proteins, which are not expected to be associated with AP-1 vesicles, are all on the left side of the plot. This demonstrates they are de-enriched relative to CLTC, despite being some of the most abundant proteins in the input (Fig. S5). The complete dataset is available in Table S1.

Although GGA2 could always be detected in the total CCV fraction, in the bead-captured samples, it was either undetectable or appeared as too few peptides for an accurate analysis. However, our Western blotting results suggest that the behavior of GGA2 is similar to that of AP-2, which is at the borderline in our proteomics dataset, indicating low levels of capture by the beads. In addition, we saw little colocalization between GGA2 and AP-1 in intact cells (see Fig. 7 and Video 7). Together, these results suggest that although there may be small amounts of

GGA2 associated with AP-1 vesicles, most of the GGA2 is associated with a different population of vesicles.

## Discussion

The AP-1 complex was first described over 35 years ago (Keen, 1987; Pearse and Robinson, 1984), but its precise function is still unclear. Genetic studies have provided some insights, but also some contradictory results, most likely because of indirect effects. The present study was an attempt to get around these problems by looking directly at AP-1 vesicles.

Using a combination of live cell imaging and single vesicle analysis, we were able to determine the time at which membrane proteins are first incorporated into AP-1 vesicles. The two AP-1 cargo proteins we investigated, CDMPR and NAGPA, were for the most part seen to leave the Golgi separately in different types of carriers. Surprisingly, it was the tubular carriers containing NAGPA, which also contained LAMP1 and thus were presumably on their way to the plasma membrane, that were decorated with AP-1 puncta. We propose that the function of the AP-1 on these tubules is to package selected membrane proteins, such as NAGPA, into CCVs to be returned to the TGN before they stray too far. The steady-state localization of NAGPA to the TGN was reached early on, and it seems likely that this localization was maintained at least in part by constant retrieval by AP-1. Some of the NAGPA traveled all the way to the plasma membrane and brought in bound nanobodies, and these NAGPA-nanobody complexes were also packaged into AP-1 vesicles.

In contrast, CDMPR took longer to reach its steady state distribution, and there was also less nanobody uptake at earlier time points, consistent with its exit from the TGN primarily in a different type of carrier, which is presumably going to endosomes rather than the plasma membrane. By 90 min, however, the CDMPR had moved out to the cell periphery and picked up nanobody. By 2 h, the majority of AP-1 vesicles contained detectable CDMPR, with nearly half of these also containing detectable nanobodies. At all time points, most of the AP-1 vesicles that contained NAGPA also contained CDMPR, but not vice versa. We think this is because of the more widespread distribution of CDMPR, which with time became increasingly colocalized with endocytosed transferrin. Thus, we propose that the function of AP-1, both on post-Golgi tubules and on early/recycling endosomes, is to sequester proteins into CCVs so that they can be returned to the late Golgi. This hypothesis is consistent with phenotypes that have been reported by ourselves and others, where proteins that localize mainly to the TGN at steady state relocate to peripheral compartments and to the

plasma membrane when AP-1 is depleted (Meyer et al., 2000, 2001; Harasaki et al., 2005; Hirst et al., 2012).

But what about our earlier study showing that acute depletion of AP-1 by knocksideways caused a greater than twofold decrease in both GGA2 and lysosomal hydrolases from the CCV-enriched fraction (Hirst et al., 2012)? Initially, we proposed two models (see Fig. 1): either that AP-1 cooperates with GGAs to facilitate forward transport of hydrolase–receptor complexes in mixed CCVs (Model 1) or that AP-1 is solely retrograde, acting indirectly to retrieve something needed by GGAs for forward transport, and only packaging empty receptors (Model 2). However, our vesicle capture experiments produced a surprising result. Instead of showing either efficient capture of both hydrolases and GGA2 or efficient capture of neither, they showed efficient capture of hydrolases but not GGA2. Our interpretation is that many of the M6P receptors packaged into vesicles by AP-1 still retain their bound hydrolases because of the pH of their environment. Dissociation of hydrolases from the two M6P receptors only occurs if the pH is below 6 (Tong and Kornfeld, 1989), and the intraluminal pH of early and recycling endosomes has been estimated to be between 6.2 and 6.5 (Yamashiro and Maxfield, 1987). Thus, we propose a third model (Fig. 10), in which hydrolase–receptor complexes are initially sorted at the TGN by GGAs, but those complexes that fail to be sorted by GGAs are then retrieved by AP-1. The GGA pathway would deliver the hydrolase–receptor complexes to an endosome with a pH below 6.0, and then the empty receptors would be retrieved by retromer and sorting nexins. Meanwhile, AP-1 would continue to retrieve hydrolase–receptor complexes from early/recycling endosomes, and would also retrieve TGN-resident proteins that had wandered away.

If AP-1 is solely retrograde, how can we explain all the other studies in mammalian cells, which support a role for AP-1 in anterograde trafficking? We think that the findings in these studies are open to different interpretations. For instance, a live cell imaging study carried out 20 years ago showed AP-1 puncta on tubules emanating from the Golgi region, containing a CIMPR

chimera, and some of these tubules appeared to fuse with transferrin-containing endosomes (Waguri et al., 2003). The authors proposed that the AP-1 was facilitating TGN-to-endosome trafficking, but what they showed was fusion of the two compartments, not the fusion of AP-1-derived vesicles with endosomes. Thus, their results are equally consistent with AP-1 trafficking in the opposite direction. Indeed, we suspect that they were visualizing the same types of tubules that we saw, although we did not see much SBP-GFP-CDMPR in the tubules. We think this is probably because of the way we made our cargo constructs. The lumenal and/or transmembrane domains of M6P receptors have been shown to play a major role in determining whether they exit the Golgi in vesicles or tubules (Chen et al., 2017), and whereas we kept the lumenal domain intact, in the earlier study, they replaced the lumenal domain with GFP.

Similarly, there are different ways of interpreting the study showing that knocking down AP-1 in cells expressing GFP-tagged CDMPR causes a dramatic increase in nanobody sulfation (Buser et al., 2022). The authors suggested that this is due to an increase in the TGN residence time of nanobody-GFP-CDMPR complexes, but their own images show that depleting AP-1 actually caused the nanobody to accumulate in peripheral endosomes, not the Golgi region. Thus, an alternative explanation is that AP-1 depletion caused the tyrosine sulfation machinery, normally resident in the TGN, to be relocated to endosomes. This machinery includes the transporter SLC35B2, and our knocksideways study suggests that the trafficking of SLC35B2 is dysregulated in AP-1 deficient cells (Hirst et al., 2012).

What about other organisms such as plants and yeast? Studies on AP-1-deficient plants showed defects in trafficking to the plasma membrane and the vacuole, leading to the idea that AP-1 is particularly important for the secretory pathway (Park et al., 2013). However, the AP-1 deficiency phenotype is far from straightforward because both plasma membrane and vacuolar proteins accumulated in the ER (Park et al., 2013; Teh et al., 2013; Wang et al., 2013), suggesting an indirect effect on ER exit. Another observation linking AP-1 to the secretory pathway in

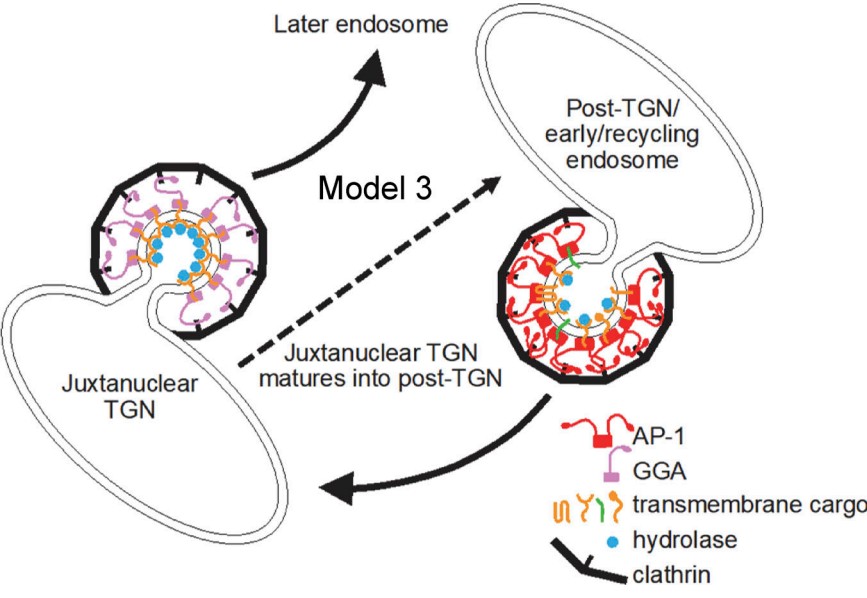

Figure 10. **A third model for AP-1 function.** This model is similar to Model 2, with AP-1 and GGAs acting independently. However, AP-1 vesicles bud from compartments (post-Golgi and early/recycling endosomes) where the pH is not low enough for hydrolases and their receptors to dissociate, and therefore the entire hydrolase–receptor complex is packaged into vesicles, together with TGN resident proteins. These vesicles return to the juxtanuclear TGN. GGAs sort hydrolases and their receptors at the juxtanuclear TGN for trafficking to a later endosome, with an intralumenal pH below 6, and empty receptors are recycled from this compartment by retromer and sorting nexins. In non-opisthokonts like plants, which do not have GGAs, we propose that the TGN-to-endosome pathway is mediated by AP-4. The juxtanuclear TGN, equivalent to the Golgi-associated TGN in plant cells, matures into the post-TGN compartment, which in plant cells would be called the Golgi-independent TGN.

plants comes from a recent high-resolution live cell imaging study on the TGN (Shimizu et al., 2021). Plant cell biologists have identified two TGN compartments: the Golgi-associated TGN (GA-TGN) and the Golgi-independent TGN (GI-TGN), with the former giving rise to the latter (Kang et al., 2011). Both forms of TGN can receive endocytosed markers, and clathrin-coated buds have been observed on both but are more abundant on the GI-TGN. The recent imaging study showed that the GI-TGN is covered with AP-1 puncta and contains proteins destined for the plasma membrane, while proteins destined for the vacuole remain in the GA-TGN, where they are thought to be sorted by AP-4. Although the authors interpreted these observations as further evidence for a role for AP-1 in the secretory pathway, once again, there are alternative explanations. We suspect that the GI-TGN in plants is essentially the same compartment as the AP-1-studded tubules, containing NAGPA and LAMP1, which we saw leaving the Golgi in HeLa cells. Several observations in plants support the hypothesis that the AP-1 associated with the GI-TGN is involved in retrieval back to the Golgi rather than in forward trafficking to the plasma membrane. First, in AP-1-deficient cells, several membrane proteins were shown to be mislocalized to the plasma membrane, including two late Golgi SNAREs: the Qa SNARE SYP41 and the Qc SNARE SYP61 (Yan et al., 2021). Second, these same two SNAREs were identified as cargo in a proteomic analysis of plant CCVs (Dahhan et al., 2022). Third, in a recent live-cell imaging study, SYP61 was shown to be packaged into AP-1 vesicles (Shimizu et al., 2021). Interestingly, the mammalian orthologues of SYP41 and SYP61, syntaxin 16 (STX16) and syntaxin 6/syntaxin 10 (STX6/STX10), respectively, were identified as AP-1 cargo both in our earlier knocksidewas study (Hirst et al., 2012) and in the present study (Fig. 9 B, arrowheads). Moreover, we showed that when AP-1 is depleted, both STX10 and STX16 become mislocalized away from the Golgi region and accumulate at the cell periphery (Hirst et al., 2012).

In yeast, a role for AP-1 in retrograde traffic was first proposed over 20 years ago, when AP-1 was identified as a hit in a screen for genes involved in intracellular retention of chitin synthase III (Valdivia et al., 2002). The authors went on to show that the Qc SNARE Tlg1, the yeast orthologue of STX6/STX10 in mammals and SYP61 in plants, is also mislocalized in AP-1-deficient cells, and they proposed that AP-1 facilitates trafficking from early endosomes to the TGN. Subsequent studies have shown that AP-1 is required for the Golgi localization of several other transmembrane proteins (Foote and Nothwehr, 2006; Day et al., 2018; Casler et al., 2022), including Tlg2, the yeast orthologue of STX16 in mammals and SYP41 in plants (Areti Pantazopoulou and Benjamin Glick, personal communication). Given the evolutionary distance between plants (Archaeplastida) and animals and yeast (both opisthokonts), we propose that the retrieval of these SNAREs is an ancient function of AP-1, most likely present in the last eukaryotic common ancestor. The role of AP-1 in SNARE localization may help to explain some of the phenotypes of AP-1 deficiency, because once SNAREs are mislocalized, there are likely to be many knock-on effects.

Thus, we would like to propose a unifying hypothesis for AP-1 function: that in all eukaryotes, its role is to transport selected proteins, including the two SNAREs, from post-Golgi compartments back to the late Golgi. The forward pathway in opisthokonts is mediated (at least in part) by GGAs, as shown in Model 3 (Fig. 10). However, GGAs are a relatively recent innovation, found only in opisthokonts, so Model 3 is not universally applicable. In plants, the TGN-to-endosome pathway is thought to be mediated by AP-4 (Fuji et al., 2016), which (like all the APs) is present in every branch of eukaryotes (Dacks and Robinson, 2017). We suggest that in some opisthokonts, the acquisition of GGAs may have made AP-4 redundant. Interestingly, many opisthokonts, including flies, worms, and most fungi, have lost all four of their AP-4 genes. Even mammals are viable without AP-4, and expression levels of AP-4 are extremely low compared with expression levels in plants (Itzhak et al., 2016, 2017; Castellana et al., 2008). Thus, a modified version of Model 3 would put AP-4 in place of GGAs for non-opisthokonts. But in any case, we propose that in all eukaryotes, the function of AP-1 is similar to the function of COPI, but acting at a later stage. Whereas COPI retrieves proteins that need to be in the ER and early Golgi, AP-1 retrieves proteins that need to be in the late Golgi. Although there are still some studies that are difficult to reconcile with an exclusively retrograde role for AP-1, we believe that this is a useful working hypothesis that will lend itself to further testing.

Finally, we think that the combination of approaches that we used in this study, single vesicle analysis and vesicle capture, should be very versatile. For instance, these two approaches could be used to characterize AP-1 vesicles in other organisms, such as yeast and plants. They could also be used to identify AP-1-dependent sorting signals in cargo proteins, such as M6P receptors and SNAREs. But this combination of approaches does not need to be limited to studies on AP-1; it could be equally applicable to studies on other transport intermediates, such as those formed using AP-3, AP-4, and COPI, all of which are components of our CCV-enriched fraction. The two approaches are complementary, because vesicle capture gives a global view of the protein composition of a population of vesicles, while single vesicle analysis reveals the heterogeneity within that population. Together, they could be valuable tools for studying many different kinds of trafficking machinery.

## Materials and methods
### Antibodies
The following antibodies were used in this study: rabbit anti-CHC17 (one made in house and one purchased from ab172958; Abcam); mouse anti-CHC17 (X22; Abcam); rabbit anti-AP1G1 (made in house); rabbit anti-GFP (one a gift from Matthew Seaman, CIMR, Cambridge, UK, and one purchased from EPR14104; Abcam); mouse anti-HaloTag (G9211; Promega); rabbit anti-CIMPR (made in house); mouse anti-AP-2 α subunits (AC1-M11; Abcam); and mouse anti-GGA2 (a gift from Doug Brooks, University of South Australia, Australia).

In addition, a new antiserum was produced against mRuby2, which was not recognized by antibodies against other red fluorescent proteins because it comes from a different organism (*Entacmaea quadricolor* instead of *Discosoma*). The entire coding sequence was expressed as a GST fusion protein, affinity-

purified using glutathione-Sepharose (GE Healthcare), and used to raise antibodies in rabbits following our established protocol (Page et al., 1999). The titer of the serum was found to be extremely high, producing very strong labeling in both immunofluorescence and Western blotting experiments when it was diluted 1:10,000, so it was not affinity purified.

## Plasmids and cells

Unless otherwise specified, all experiments were carried out on HeLa M cells (ECACC). Initially, we tried tagging endogenous *AP1G1* by gene editing using homology-directed repair to insert mRuby2 into the hinge region. As shown in Fig. S1 A, this worked in principle, but expression was always much lower than that of wild-type AP1G1 in the parental cells. We assume this is because of the extreme aneuploidy of HeLa cells so that we were only tagging one to three copies of the *AP1G1* gene and often introducing indels into the others.

Because we needed tagged *AP1G1* to be expressed at endogenous levels, we used a different strategy: to introduce the tagged *AP1G1* by retroviral transduction and then to knock out the endogenous gene. First, we used Gibson assembly to make a construct in a modified version of the retroviral vector pLXIN (a gift from Andrew Peden, University of Sheffield, Sheffield, UK), which has a moderate LTR promoter. The construct consisted of the mouse *AP1G1* cDNA (Robinson, 1990) with mRuby2 inserted between amino acids 667 and 668. We also introduced four silent mutations near the beginning of the coding sequence to make the cDNA resistant to our guide RNAs. The resulting plasmid, pAP1G1-mRuby2, was mixed with packaging plasmids pMD.Gag-Pol and pMD.VSVG in a ratio of 10:7:3 and transfected into HEK293 cells (ECACC) using TransIT-293 (Mirus Bio LLC), according to the manufacturer's instructions. Viral supernatants were harvested after 48 h, filtered through a 0.45-µm filter, and applied directly to the target cells (Hirst et al., 2021). Antibiotic selection for stable expression in 500 µg/ml G418 was initiated 48 h after transduction. We then used gene editing to disrupt the endogenous *AP1G1* gene, targeting the sequence 5′-CCATCAGAT TGCGGGAGCTGATC-3′. Guide RNAs were synthesized and cloned into pX330 (Addgene), and the plasmid was transfected into the cells expressing tagged *AP1G1* using a TransIT-HeLaMONSTER kit (Mirus Bio LLC), according to the manufacturer's instructions. Clonal cell lines were isolated and assayed by Western blotting (see below) for loss of the endogenous protein and expression of the tagged protein at near-endogenous levels. The clonal cell line A5 was used as the basis for all further experiments.

To make use of the RUSH system (Boncompain et al., 2012), we used the construct pDCGx4 (Chen et al., 2017), which encodes a myc-tagged Hook construct, downstream from a strong CMV promoter, in the retroviral vector pQCXIP. The A5 cell line was transduced as described above and positive cells were selected by puromycin resistance using a concentration of 1 µg/ml. This (non-clonal) stably transduced cell population was called AH. Cells were routinely screened by immunofluorescence for Hook expression using the mouse anti-myc monoclonal 9E10 (Abcam). Expression was normally very stable, but occasionally the expression became more heterogeneous. When this happened, we transduced the cells again using the same method.

Cargo constructs were made using the coding sequences of CDMPR, NAGPA, and LAMP1. The original CDMPR construct, with a streptavidin-binding peptide followed by EGFP, is described in Chen et al. (2017). It was moved into the same modified pLXIN retroviral vector that was used for *AP1G1*. The resulting plasmid, pVGM, was transduced into AH cells to create AHGM cells, which were sorted by flow cytometry using an Influx cell sorter (BD Biosciences). The retroviral constructs VGN and VGL were made by replacing the CDMPR coding sequence with the coding sequences (just after the signal peptide) of NAGPA (cDNA purchased from Stratech, HG20591-UT-SIB) and LAMP1 (Chen et al., 2017), respectively. Again, after transduction into AH cells, the resulting AHGN and AHGL cells were sorted by flow cytometry. HaloTag-conjugated versions of CDMPR and NAGPA, pVOM and pVOM, were also constructed, replacing EGFP with HaloTag (Addgene). These plasmids were then stably transduced into AHGM cells to generate AHGMOM and AHGMON cell populations, which were sorted by flow cytometry. pVOM was also transiently transfected into AHGL cells for live imaging. HaloTag was visualized with the dye Janelia Fluor 646 (Promega) at a concentration of 200 nM.

A HaloTag-conjugated nanobody against GFP was made for uptake experiments, following the protocol of Buser et al. (2018). A plasmid encoding mCherry-tagged camelid anti-GFP, pET24a-VHH-mCherry, was obtained from Addgene, and HaloTag was then inserted in place of mCherry to produce pET24a-VHH-HaloTag. This plasmid was transformed into BL21(DE3) cells (Thermo Fisher Scientific) and purified using nickel Sepharose as described (Buser et al., 2018). The resulting nanobody was added to the culture medium at a final concentration of 5–10 µg/ml, after first titrating it to find the optimum concentration for specific uptake.

Two other constructs were used for transient transfections, gadkin and GGA2. The gadkin cDNA (Hirst et al., 2015), followed by HaloTag, was inserted into pEBFP-N1 (Clontech) cut with EcoRI and NotI (so the BFP dropped out), using Gibson assembly. Full-length GGA2 cDNA (Hirst et al., 2012) was internally tagged with mClover3 and inserted into modified pLXIN using Gibson assembly.

For imaging fixed cells, the cells were grown on coverslips, fixed in 3.7% formaldehyde in PBS for 10 min, and mounted onto slides. Widefield images were captured on a Zeiss Axio Imager II microscope using a 63×/1.4 NA oil immersion objective and AxioCam 506 camera.

## Lattice SIM microscopy

For live-cell imaging, cells were grown in MatTek dishes and imaged in an Elyra 7 with Lattice SIM2 microscope (Zeiss) equipped with an environmental chamber (temperature controlled at 37°C, humidified 5% $CO_2$ atmosphere), two PCO.edge sCMOS version 4.2 (CL HS) cameras (PCO), solid-state diode continuous-wave lasers, and a Zeiss Plan- Apochromat 63×/1.4 Oil DIC M27, all under the control of ZEN black software (Zeiss). Biotin (400 µM), Janelia Fluor 646 (200 nM), and/or Alexa Fluor 647-conjugated human transferrin (25 µg/ml; Invitrogen) were added to the cells for some experiments.

## Western blotting

Western blotting was carried out as previously described (Hirst et al., 2021), using the antibodies listed above. HRP-conjugated secondary antibodies were purchased from Sigma-Aldrich. For probing blots of captured AP-1 vesicles, we used either HRP-conjugated protein A or HRP-conjugated protein G (both from Cell Signaling) because of the high background from the immunoglobulin bands when we used secondary antibodies. HRP was detected using ECL Prime reagent (Amersham). In most cases, the blots were then exposed to x-ray film; however, the images in Fig. 8 C and Fig. S4 were acquired using a BioRad ChemiDoc imaging system and FIJI software.

## CCV isolation and single vesicle analysis (SVA)

Our protocol for isolating CCVs from HeLa cells is described in Borner et al. (2006). Briefly, it involves homogenizing the cells, centrifuging at a relatively low speed to remove unbroken cells and large debris, and then centrifuging the supernatant at high speed to pellet organelles, vesicles, and smaller particles such as ribosomes. The resuspended pellet is mixed with an equal volume of 12.5% Ficoll/12.5% sucrose and centrifuged at a moderate speed to pellet larger structures such as organelles, and then the supernatant is diluted and centrifuged at a higher speed to pellet smaller structures, with the final centrifugation step carried out at 50,000 RPM. Pellets were then resuspended in 20 µl buffer A. For some experiments, the cells were pretreated before starting the isolation procedure. Biotin was added to cells at a concentration of 400 µM for 30, 40, or 120 min. In some of these experiments, the HaloTag-conjugated nanobody described above was added for the final 30 min and Janelia Fluor 646 was added to the low-speed supernatant. In other experiments, Alexa Fluor 647-conjugated human transferrin (Invitrogen) was added to the cells for 1 h at a concentration of 25 µg/ml. For a typical experiment, we used 12 confluent 150-cm dishes and began by reducing the amount of medium to 10 ml per dish. The dishes were then divided into four sets of three, with three dishes for every time point. The dishes were then returned to the incubator to equilibrate. We staggered the time points to process one set of dishes every 20 min. Thus, the timing for a typical experiment, in which cells were treated with biotin for 0, 30, 40, and 120 min (sets 1–4) and also fed transferrin, is shown below. Note that every set of dishes was incubated with transferrin for exactly 1 h and then harvested immediately.

Time 0: Transferrin to set 1, biotin to set 4
20 min: Transferrin to set 2
40 min: Transferrin to set 3
50 min: Biotin to set 2
60 min: Harvest set 1, biotin to set 3, transferrin to set 4
80 min: Harvest set 2
100 min: Harvest set 3
120 min: Harvest set 4

The slides used for SVA were precoated with 0.1 µm fluorescent beads (Invitrogen TetraSpeck microspheres) to act as a fiducial marker because the channels needed to be aligned for every image. The beads were diluted 1:300 in 1 mg/ml BSA in water (the BSA ensured that the beads adhered to the slides). An ~1 cm–diameter circle was drawn on the bottom of microscope slides and 2 µl of the bead suspension spread over this circle on the other side. The slides were then allowed to dry on a 37° dry block heater. Resuspended CCV-containing pellets were diluted 1:2, 1:4, and 1:8, and 2 µl were aliquoted onto the slides on top of the beads. 2 µl mounting medium (Invitrogen ProLong Diamond Antifade Mountant) was aliquoted onto coverslips, and the coverslips were placed over the circles, gently pressed down, and allowed to dry. The beads fluoresced in four wavelengths (green, red, far-red, and blue), while the CCV-enriched samples only fluoresced in three (green, red, and far-red), so it was possible to identify the beads, which were otherwise difficult to distinguish from CCVs as they are of similar size and brightness.

To visualize clathrin on the slides by immunofluorescence, bead-coated slides were prepared as described, and then circles were drawn on the top of the slide with an ImmEdge pen (Vector Laboratories) after the beads had dried. Then, after applying the resuspended pellet, the slides were fixed with 3% paraformaldehyde in PBS, blocked with 1 mg/ml BSA in PBS (PBS/BSA), and permeabilized with 0.1% Triton X-100. Antibody incubations were carried out in PBS/BSA for 1 h, with washes in between. The primary antibody was X22 (see above); the secondary antibody was Alexa Fluor 499-conjugated donkey anti-mouse IgG (Thermo Fisher Scientific).

Widefield images of the vesicles were captured as above, but by using a 100×/1.4 objective. At least 10 images were collected for each condition using varying dilutions of the pellet. Each image contained anywhere from 1,000 to 10,000 discrete spots. Three biological repeats were carried out for each experiment. A newly written script was used for image analysis (see below).

## Quantification of SVA

To analyze our dataset, we utilized a Python-based pipeline. Raw microscopy data in CZI format were loaded from a designated directory, with each set containing multiple channels, including fluorescence channels for different markers and a fiducial bead channel. Employing fiducial beads as references, a homographic alignment was performed, using the green fluorescence channel as the primary reference. The blue and red channels were aligned to the green channel, while the far-red was aligned to the red channel, utilizing OpenCV's findTransformECC and warpPerspective functions. Punctate structures across channels were discerned using the Laplacian of Gaussian (LoG) filter from the skimage library. For each identified structure, a region of interest (ROI) was demarcated based on its radius, and the mean fluorescence intensity within this ROI was calculated for all channels. Leveraging single-channel controls (i.e., "monochrome CCVs" from cells expressing mRuby2-AP-1 only, SBP-GFP-CDMPR only, or wild-type cells incubated with far-red transferrin), thresholds were set to binarize data, determining vesicle cargo and adaptor presence. Notably, fiducial beads, consistent across channels, were excluded from the analysis. After processing, results were aggregated and exported as CSV files. This methodology facilitated robust and reproducible vesicle categorization based on the presence of machinery or cargo. The script can be downloaded at https://github.com/GershlickLab/SVA/.

## Vesicle capture

AP-1-positive vesicles were captured from A5 cells using the mRuby2 antiserum described above and protein A-coated magnetic Dynabeads (Invitrogen). The beads were first incubated with either mRuby2 antiserum or the preimmune serum from the same rabbit using 10 µl serum for 50 µl beads, following the manufacturer's instructions. After washing with PBS containing 0.2% Tween 20, the beads were washed three times with PBS alone (because the presence of the detergent might extract membrane proteins from the vesicles). A CCV-enriched preparation was carried out using our standard protocol (see above), but before the final centrifugation step, two volumes of 0.33 M MOPS, pH 7.4, were added to the supernatant to raise the pH and facilitate binding. The CCV-enriched supernatant was then rotated with the beads for 1 h at room temperature and washed with a 2:1 mixture of 0.33 M MOPS and Buffer A. The washed beads were then either prepared for electron microscopy (see below) or eluted with SDS-containing sample buffer for gel electrophoresis, usually followed by Western blotting (see above).

For electron microscopy, the beads were fixed in 2.5% glutaraldehyde in 0.1 M cacodylate buffer, pH 7.4, postfixed with 1% osmium tetroxide in 0.1 M cacodylate buffer, pH 7.4, stained en bloc with aqueous uranyl acetate, serially dehydrated with ethanol, and embedded in Agar 100 resin (Agar Scientific). Sections were stained with uranyl acetate and lead citrate before being imaged on a Tecnai Spirit transmission electron microscope at an operating voltage of 80 kV.

## Proteomics

To quantify bead-captured vs. total CCV-enriched proteins, A5 cells were metabolically labeled in either SILAC heavy or SILAC light medium for at least 2 wk (Ong et al., 2002). The labeling was performed in SILAC RPMI 1640 medium (89984; Thermo Fisher Scientific), supplemented with 10% (vol/vol) dialyzed fetal calf serum (10,000 MW cut-off; Invitrogen) and either "Heavy" amino acids (50 µg/ml $^{13}C_6$, $^{15}N_4$ Arginine HCl and 100 µg/ml $^{13}C_6$, $^{15}N_2$ Lysine 2HCl; Cambridge Isotope Laboratories) or the equivalent "Light" amino acids. CCV-enriched preparations were carried out in parallel on both sets of cells (12 confluent 150-mm dishes for each). Before the final centrifugation step, the preparation from the cells grown in a heavy medium was incubated with anti-mRuby2 coupled to magnetic beads to capture AP-1 vesicles, as described above. The preparation from the cells grown in a light medium was centrifuged one last time at 50,000 RPM to obtain an input fraction for reference. The two samples were made up to the same volume in SDS-containing sample buffer, mixed, loaded in a single lane onto a preparative 1.5-mm 10% acrylamide gel, and run so that the sample separated into a 5-cm strip. The gel was then washed, stained with Coomassie blue, and cut into 10 slices. Proteins were reduced with 10 µM DTT (Sigma-Aldrich), alkylated with 50 µM iodoacetamide (Sigma-Aldrich), and in-gel digested with 10 ng/µl trypsin (Roche).

Digested peptides were collected in 0.5 ml tubes (Protein LoBind; Eppendorf) and dried almost to completion. Samples were resuspended in 20 µl solvent (3% MeCN, 0.1% TFA) and 6 µl were analyzed by LC-MSMS using a Thermo Fusion Lumos mass spectrometer (Thermo Fischer Scientific) equipped with an EASYspray source and coupled to an RSLC3000 nano UPLC (Thermo Fischer Scientific). Peptides were fractionated using a 50-cm C18 PepMap EASYspray column maintained at 40°C with a solvent flow rate of 300 nl/min. A gradient was formed using solvent A (0.1% formic acid) and solvent B (80% acetonitrile, 0.1% formic acid), raising solvent B from 7 to 37% by 88 min, followed by a 2-min wash using 95% solvent B. MS spectra were acquired at 120,000 resolution between m/z 350 and 1,500 with MSMS spectra acquired in the ion trap following HCD activation. Data were processed in Maxquant 2.2.0.0 with carbamidomethyl (C) set as a fixed modification, and oxidation (M) and acetyl (protein N-terminus) set as variable modifications. Data were searched against a Uniprot Homo Sapien database (downloaded 06/12/22) and a database of common contaminants.

To compare the heavy (H) and light (L) datasets, mass spectrometry raw files were processed in MaxQuant Version X (Cox and Mann, 2008). Downstream data analysis was performed on the "proteinGroups.txt" output file, and all data transformation, filtering, and statistical analyses were performed in Perseus software version 1.6.2.3 (Tyanova et al., 2016). For statistical analysis of the four datasets, they were first subjected to a standard filter for reverse hits, matches based on modified peptides only, and common contaminants. Then the rows were filtered to include only those proteins that had at least three valid H/L ratios (i.e., were quantified in three out of the four datasets), leaving 632 proteins. Next, for each of the four experiments, the H/L ratios were normalized by dividing by the median H/L ratio for that experiment, to make them comparable, and the normalized H/L ratios were then $\log_2$-transformed. We then performed a one-sample $t$ test (two-tailed) to compare the normalized $\log_2$ ratios for each protein to a hypothetical ratio defined as the mean normalized $\log_2$ ratio for CLTC (0.0976484). The rationale for using CLTC is that it is the most abundant protein in both the CCV input sample and the AP-1 IP (see Fig. S5). CLTC is known to be associated with AP-1 vesicles, but not all the CLTC is expected to associate with AP-1 vesicles, as the preparation also contains AP-2 vesicles and probably reassembled empty clathrin coats as well. Therefore, we would expect any protein with a ratio greater than CLTC to be a potential component of AP-1 vesicles. In Fig. 9 we plot the results of the $t$ test, with the $t$ test difference ($\log_2$ fold enrichment over CLTC in the IP) on the x-axis and the unadjusted $-\log_{10}$ P value on the y-axis. We used the Benjamini-Hochberg procedure to correct for multiple hypothesis testing and report the false discovery rate (FDR)-adjusted q values in Table S1.

## Online supplemental material

Fig. S1 shows our attempt to tag the AP-1 γ subunit using gene editing, how we selected for cells co-expressing mRuby2-tagged AP-1 and GFP-tagged membrane proteins using flow cytometry, and recruitment of AP-1 onto LAMP1-containing tubules leaving the Golgi. Fig. S2 shows an example of a widefield image used for single vesicle analysis. Fig. S3 shows results from all 18 of the experiments used for single vesicle analysis and includes AP-1-negative puncta as well as AP-1-positive puncta. Fig. S4 shows that GFP-tagged constructs were expressed at higher levels than

HaloTag constructs, and that co-expressed GFP-CDMPR and HaloTag-CDMPR were almost complete colocalized in intact cells. Fig. S5 shows the protein rank plots for both bead-captured vesicles and the total CCV-enriched fraction. Video 1 shows LAMP1 and NAGPA leaving the Golgi in the same tubules. Video 2 shows AP-1 puncta on NAGPA-positive tubules leaving the Golgi. Video 3 shows AP-1 puncta on NAGPA-positive tubules leaving the Golgi, and not on CDMPR-positive vesicular profiles. Video 4 shows AP-1 on LAMP1-positive tubules leaving the Golgi. Video 5 shows AP-1 puncta on both NAGPA-positive tubules leaving the Golgi and tubules containing endocytosed transferrin. Video 6 shows the same cell shown in Video 5 to highlight the transferrin-positive tubules. Video 7 shows that AP-1 and GGA2 are mostly non-coincident. Table S1 contains all the mass spectrometry data on bead-captured proteins and the input reference from SILAC-labeled cells. Table S2 contains the mass spectrometry data on bead-captured proteins from unlabeled cells, comparing preimmune serum with anti-mRuby2.

### Data availability

The data underlying the main and supplemental figures are available in the published article and its online supplemental material. The mass spectrometry proteomics data used to generate Tables S1 and S2 have been deposited to the ProteomeXchange Consortium via the PRIDE partner repository with the dataset identifier PXD050449.

### Acknowledgments

We thank Reiner Schulte, Gabriela Grondys-Kotarba, and Chiara Cossetti for cell sorting; Matthew Gratian and Mark Bowen for help with light microscopy; Nick Bright for help with electron microscopy; John Suberu for preparing samples for mass spectrometry; and Paul Luzio for reading the manuscript and for helpful discussions.

This work was funded by Wellcome grants 214272 to M.S. Robinson and 212892 to the Cambridge Institute for Medical Research. D.C. Gershlick is funded by a Biotechnology and Biological Sciences Research Council project grant (BB/W005905/1) and a Wellcome Trust/Royal Society Sir Henry Dale Fellowship (210481).

Author contributions: M.S. Robinson conceived of the project and designed and performed most of the experiments. D.C. Gershlick wrote the script for single vesicle analysis and performed the live cell imaging. A. Sanger designed and performed the immunofluorescence assay on isolated vesicles. R. Antrobus performed the mass spectrometry and the initial analysis of the proteomics data. A.K. Davies carried out further analyses of the proteomics data. The manuscript was written by M.S. Robinson, with contributions from the other co-authors.

Disclosures: The authors declare no competing interests exist.

Submitted: 27 October 2023

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

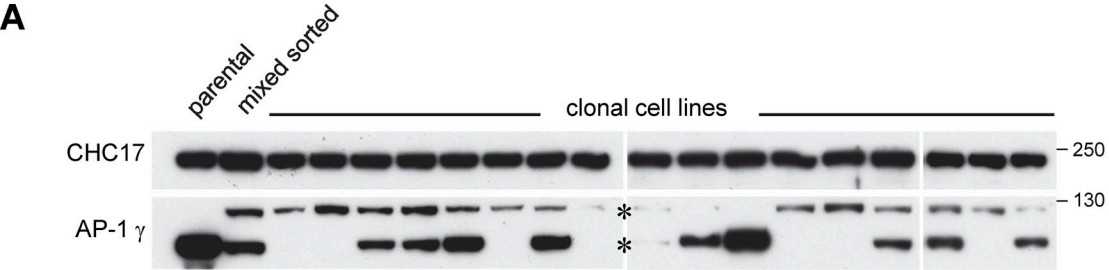

**A**

parental

mixed sorted

clonal cell lines

CHC17

AP-1 γ

— 250

— 130

**B**

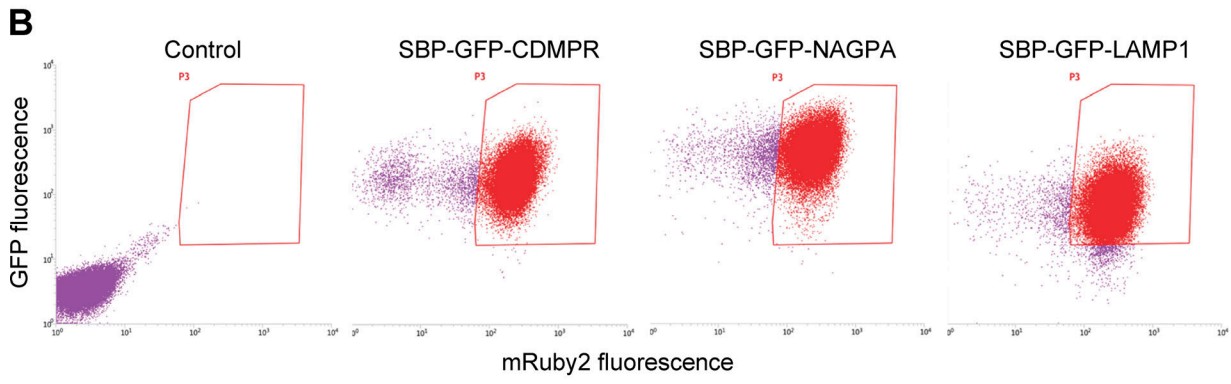

Control    SBP-GFP-CDMPR    SBP-GFP-NAGPA    SBP-GFP-LAMP1

GFP fluorescence

mRuby2 fluorescence

**C**

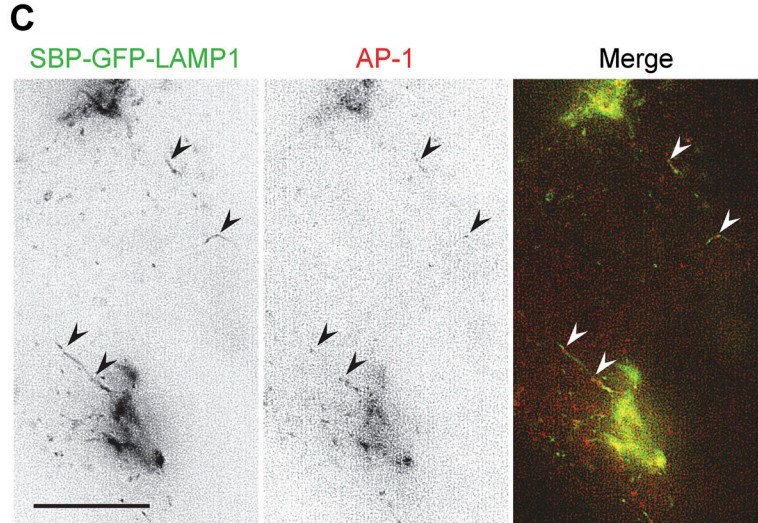

SBP-GFP-LAMP1    AP-1    Merge

Figure S1.    **Generation and characterization of cell lines and cell populations. (A)** Initially, we attempted to tag the AP-1 γ subunit with mRuby2 using gene editing. Positive cells were sorted by flow cytometry and then individual clonal cell lines were isolated and analyzed by Western blotting. The blots show that although the tagging was successful, expression levels for the tagged protein (upper band) were always much lower than that of the wild-type protein in the parental HeLa cells. This is most likely because the gene we tagged, *AP1G1*, is present in multiple copies, and only very few were tagged in each cell line, while the others were frequently disrupted by indels. **(B)** After generating a clonal cell line in which the tagged AP-1 γ subunit was introduced by retroviral transduction and the endogenous gene was then deleted using gene editing, we added additional tagged membrane proteins using retroviral transduction. Cells were selected by flow cytometry and then routinely sorted by flow cytometry before scaling up for CCV isolation experiments. The dot plots show that there was some loss of mRuby2-tagged AP-1 γ with time and that although the three membrane proteins were under the control of the same LTR promoter and were sorted using the same gates, there were inherent differences in expression levels, with SBP-GFP-NAGPA being most strongly expressed, followed by SBP-GFP-CDMPR and then SBP-GFP-LAMP1. **(C)** Frames from Video 4, showing cells co-expressing SBP-HaloTag-NAGPA (red) and SBP-GFP-LAMP1 (green), treated with biotin for 28 min. Some of the LAMP1-containing tubules are decorated with AP-1, indicating that the presence of AP-1 on these tubules is not simply a consequence of overexpressing the AP-1 cargo protein NAGPA. Scale bar: 10 μm. Source data are available for this figure: SourceData FS1.

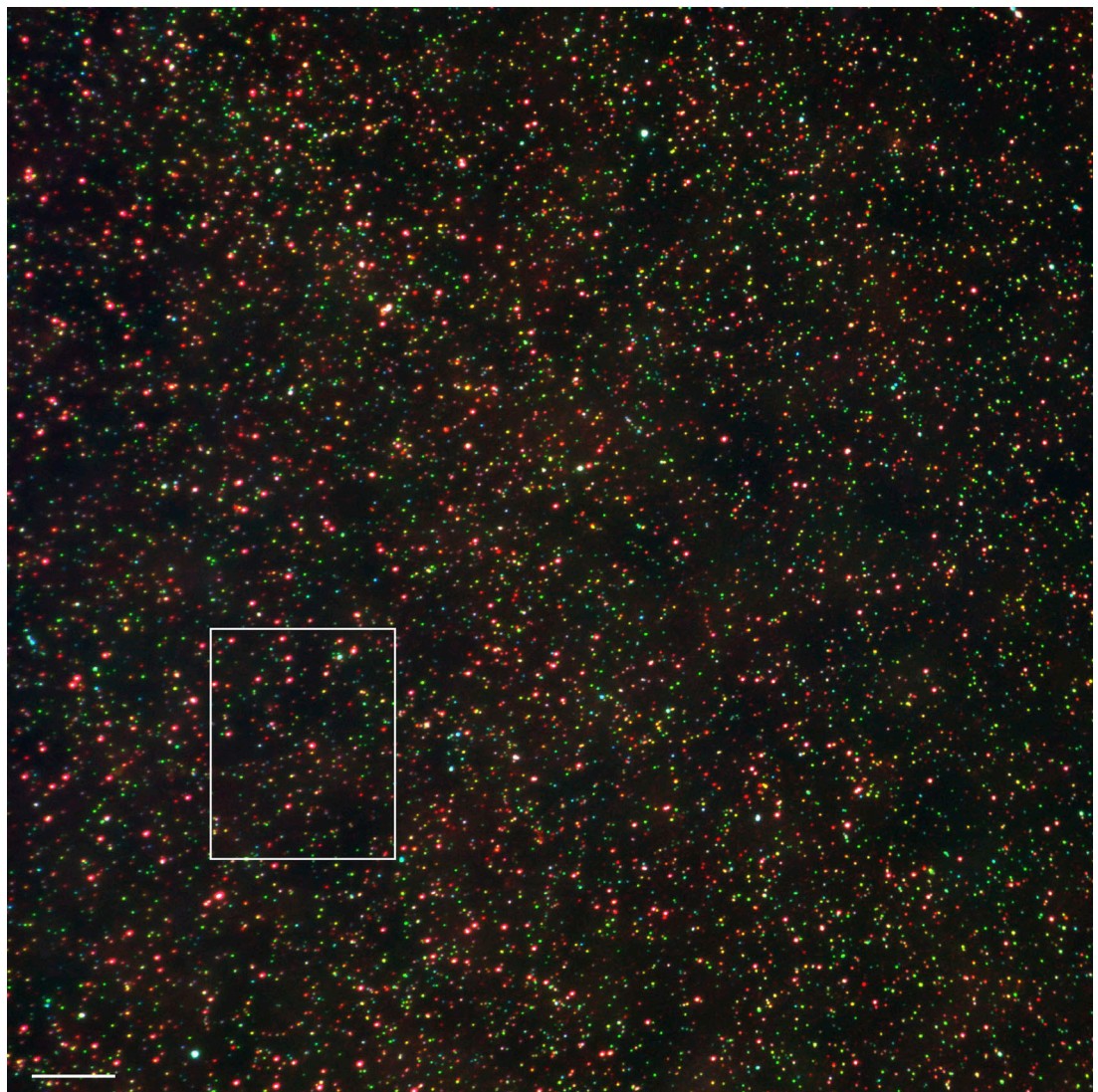

Figure S2.   **Example of a widefield image used for single vesicle analysis.** The region shown in Fig. 5 B is indicated. Scale bar: 10 µm.

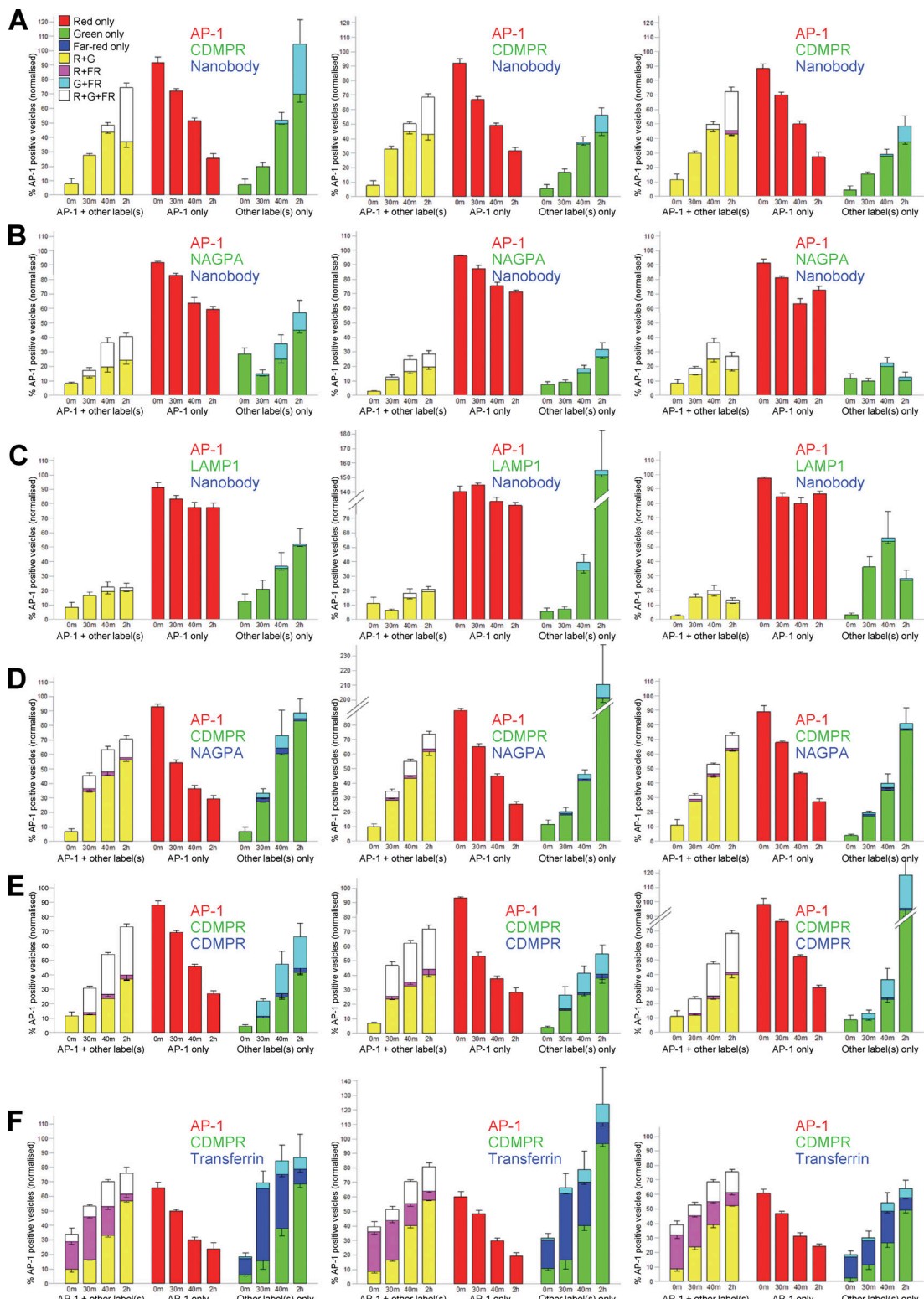

Figure S3.  **Results from all 18 of the experiments used for single vesicle analysis.** Three biological repeats were carried out for each condition. All of the cells used in these experiments expressed mRuby2-tagged AP-1 (red) but their GFP-tagged and far-red proteins were different (shown in green and blue, respectively). The key in the upper left shows the different combinations of labels, based on how they appear in the images (e.g., R+G for red and green only, shown as yellow; R+FR for red and far-red only, shown as magenta, etc.). On the actual graphs, the right-hand "AP-1 + other label(s)" portions (R+G/yellow, R+FR/magenta, and R+G+FR/white) were used to generate Figs. 5 and 6. The middle portion of each graph (red) shows the percentage of vesicles that were positive for AP-1 only. The right-hand portions (G, FR, G+FR/cyan) show vesicles that were negative for AP-1 but positive for other fluorescent proteins. These were normalized to total AP-1-containing vesicles, and thus there are some instances where the percentage is >100. Although the AP-1-containing vesicles were very consistent between experiments, there was much more variability in non-AP-1 puncta. The error bars indicate the standard deviation.

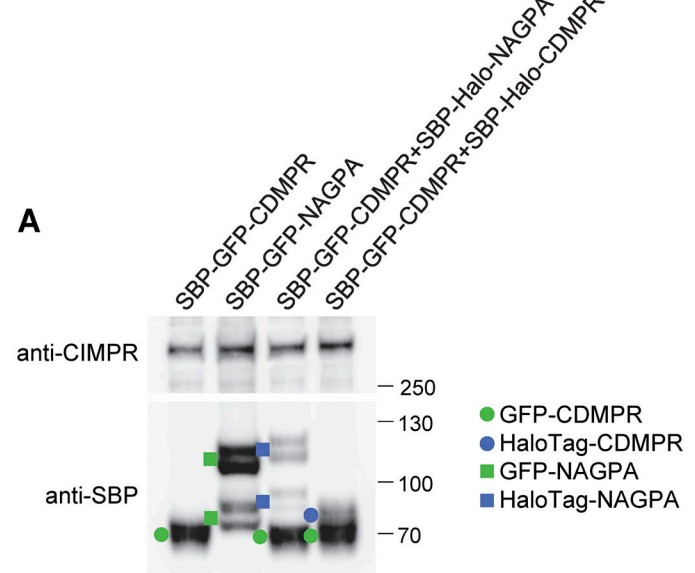

**A**

anti-CIMPR

anti-SBP

— 250
— 130
— 100
— 70

● GFP-CDMPR
● HaloTag-CDMPR
■ GFP-NAGPA
■ HaloTag-NAGPA

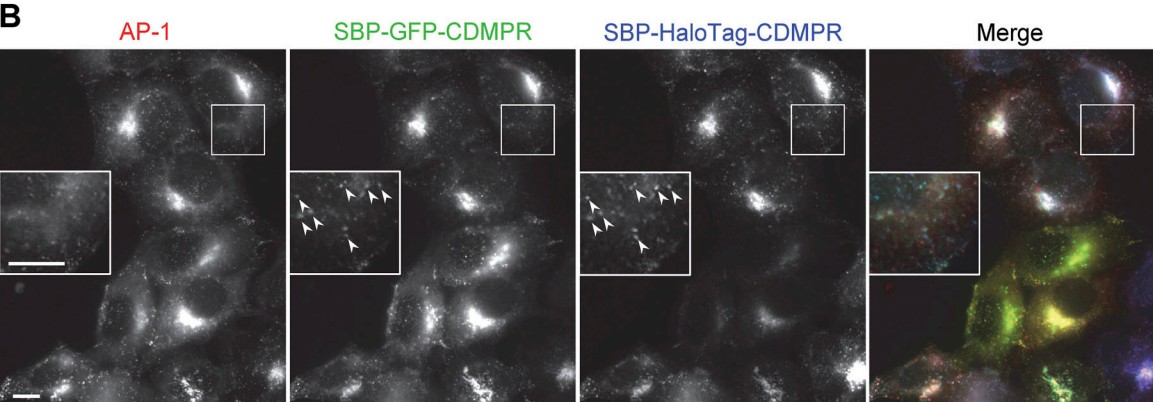

**B** AP-1 SBP-GFP-CDMPR SBP-HaloTag-CDMPR Merge

Figure S4. **Further characterization of AP-1 cargo constructs. (A)** Western blots comparing expression levels of GFP- and HaloTag-constructs. The cells were all incubated with biotin for 40 min. In every case, the constructs labeled with GFP are expressed at ~threefold higher levels than the same constructs labeled with HaloTag. **(B)** Fluorescent images of cells coexpressing GFP-CDMPR and HaloTag-CDMPR. Although there is variability in relative expression levels when different cells are compared, the fine details are virtually identical (insets). Scale bars: 10 µm. Source data are available for this figure: SourceData FS4.

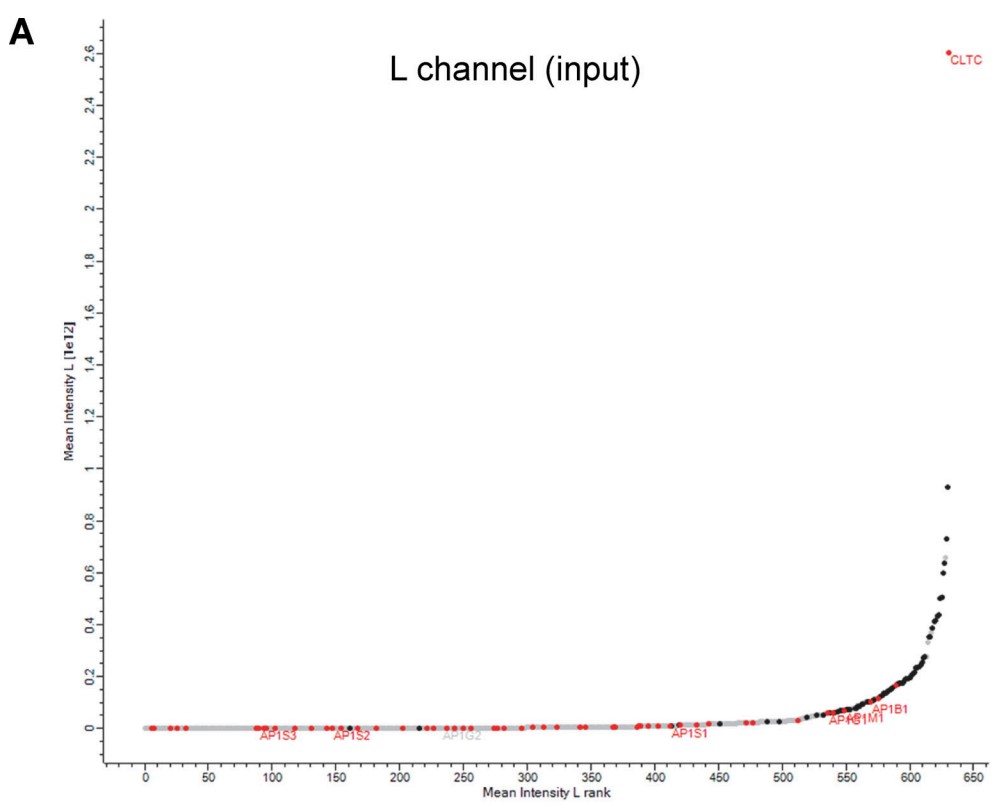

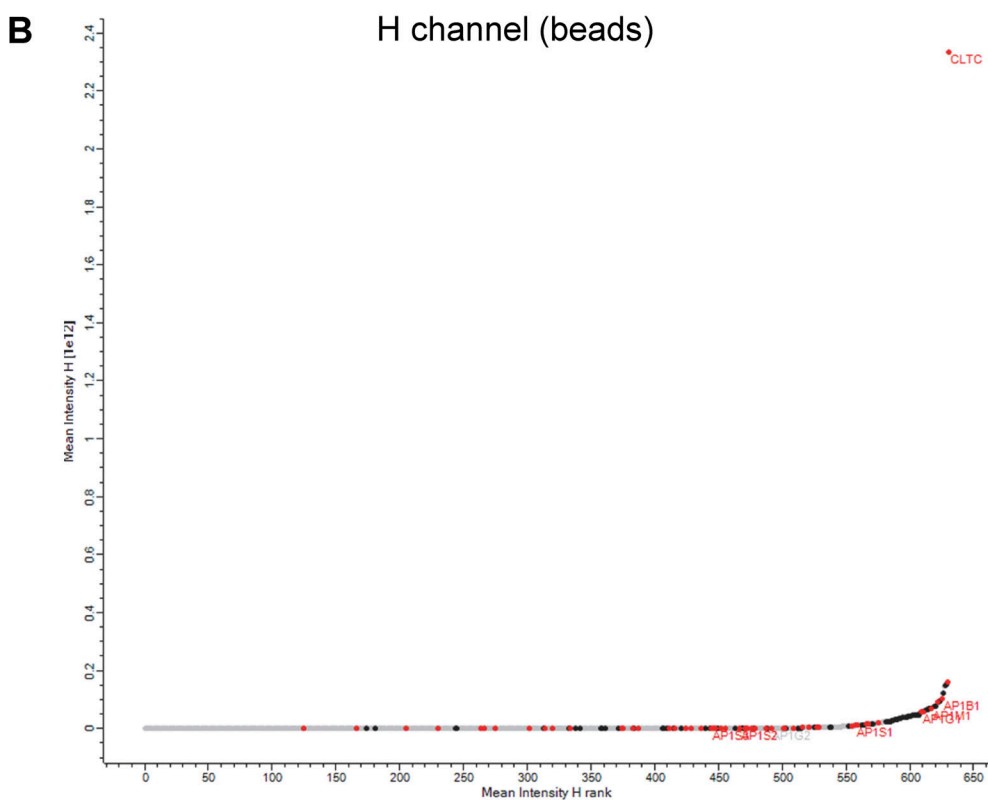

Figure S5. **Protein rank plots.** The graphs show the mean rank (0 is lowest intensity) versus the mean $\log_{10}$ intensity for each of the 632 proteins analyzed in Fig. 9. The L (light) channel is the input, i.e., the total CCV-enriched fraction, while the H (heavy) channel is the bead-captured proteins. CHC17 (CLTC) is the most abundant protein in both channels and ribosomal proteins (black) are also abundant in both channels. However, AP-1, as well as AP-1-associated proteins identified in our previous knocksideways study (red) (Hirst et al., 2012), shift up the rank distribution in the heavy channel. The L (light) channel (A) is the input, i.e., the total CCV-enriched fraction, while the H (heavy) channel (B) is the bead-captured proteins.

Video 1.   **Cells coexpressing SBP-GFP-LAMP1 (green) and SBP-HaloTag-NAGPA (red) were treated with biotin for 20 min.** LAMP1 and NAGPA can be seen leaving the Golgi in the same tubules (arrowheads). Playback speed, 4x.

Video 2.   **Cell coexpressing mRuby2-AP-1 (red) and SBP-GFP-NAGPA (green) were treated with biotin for 33.5 min.** AP-1 puncta can be seen forming on NAGPA-positive tubules (arrowheads). Playback speed, 4x.

Video 3.   **Cell coexpressing mRuby2-AP-1 (red), SBP-HaloTag-NAGPA (green), and SBP-GFP-CDMPR (blue), and treated with biotin for 27 min.** AP-1 puncta can be seen forming on NAGPA-positive tubules. Small amounts of CDMPR are also in the tubules, but most of the CDMPR leaves in vesicular carriers. The arrowhead on the left shows an AP-1-positive NAGPA tubule being pulled out from the Golgi; the one on the right shows a fast-moving AP-1-positive NAGPA tubule. Playback speed, 28x.

Video 4.   **Cell co-expressing mRuby2-AP-1 (red) and SBP-GFP-LAMP1 (green), treated with biotin for 28 min.** AP-1 puncta can be seen on LAMP1-positive tubules, indicating that it binds to these tubules whether or not the cells are overexpressing an AP-1 cargo protein like NAGPA. The arrowheads show such both Golgi-associated and free-moving tubules. Playback speed, 9x.

Video 5.   **Cell coexpressing mRuby2-AP-1 (red) and SBP-GFP-NAGPA (green), fed far-red transferrin (blue) for 1 h, and treated with biotin for 27 min.** AP-1 puncta can be seen on both NAGPA-positive tubules and transferrin-positive tubules (arrowheads). In the region indicated with the upper arrowhead, the two types of tubules appear possibly to merge. Playback speed, 28x.

Video 6.   **Cell expressing mRuby2-AP-1 (red), fed far-red transferrin (green) for 1 h.** This is the same cell shown in Video 5, but with NAGPA omitted, transferrin shown in green, and AP-1 in red, to highlight how the more peripheral transferrin-containing tubules are frequently decorated with AP-1 puncta. Playback speed, 28x.

Video 7.   **Cells coexpressing mRuby2-tagged AP-1 (red) and GFP-tagged GGA2 (green).** The puncta are mostly non-coincident. Playback speed, 16x.

**Provided online are Table S1 and Table S2. Table S1 shows mass spectrometry data on bead-captured proteins and CCV input reference from SILAC-labeled cells. Table S2 shows mass spectrometry data on bead-captured proteins from unlabeled cells expressing mRuby2-tagged AP-1 γ.**

