## [Peer Review File · The Journal of Cell Biology]

The role of the AP-1 adaptor complex in outgoing and incoming membrane traffic

Margaret Robinson, Robin Antrobus, Anneri Sanger, Alexandra Davies, and David Gershlick

Corresponding Author(s): Margaret Robinson, University of Cambridge

Review Timeline:

Submission Date:	2023-10-27
Editorial Decision:	2023-11-21
Revision Received:	2024-02-17
Editorial Decision:	2024-02-20
Revision Received:	2024-03-11

Monitoring Editor: Elizabeth Miller

Scientific Editor: Andrea Marat

Transaction Report:

DOI: <https://doi.org/10.1083/jcb.202310071>

November 21, 2023

Re: JCB manuscript #202310071

Prof. Margaret S. Robinson
University of Cambridge
CIMR
Cambridge CB2 0XY
United Kingdom

Dear Prof. Robinson,

Thank you for submitting your manuscript entitled "The role of the AP-1 adaptor complex in outgoing and incoming membrane traffic". The manuscript was assessed by expert reviewers, whose comments are appended to this letter. We invite you to submit a revision if you can address the reviewers' key concerns, as outlined here.

As you will see, the reviewers appreciate that your study provides important insight towards resolving the function of AP-1, and as such we agree it should be of high interest for the readership of JCB. They have provided constructive feedback to further improve your study, much of which can be addressed through edits to your text and figures, as well as some minor experimental revisions. Regarding the overexpression concern of reviewer 2, we would welcome data to this effect, but if it is not possible, please temper your interpretation accordingly.

GENERAL GUIDELINES:

Text limits: Character count for an Article is < 40,000, not including spaces. Count includes title page, abstract, introduction, results, discussion, and acknowledgments. Count does not include materials and methods, figure legends, references, tables, or supplemental legends.

Figures: Articles may have up to 10 main text figures. Figures must be prepared according to the policies outlined in our Instructions to Authors, under Data Presentation, <https://jcb.rupress.org/site/misc/ifora.xhtml>. All figures in accepted manuscripts will be screened prior to publication.

Supplemental information: There are strict limits on the allowable amount of supplemental data. Articles may have up to 5 supplemental figures. Up to 10 supplemental videos or flash animations are allowed. A summary of all supplemental material should appear at the end of the Materials and methods section.

Please note that JCB now requires authors to submit Source Data used to generate figures containing gels and Western blots with all revised manuscripts. This Source Data consists of fully uncropped and unprocessed images for each gel/blot displayed in the main and supplemental figures. Since your paper includes cropped gel and/or blot images, please be sure to provide one Source Data file for each figure that contains gels and/or blots along with your revised manuscript files. File names for Source Data figures should be alphanumeric without any spaces or special characters (i.e., SourceDataF#, where F# refers to the associated main figure number or SourceDataFS# for those associated with Supplementary figures). The lanes of the gels/blots should be labeled as they are in the associated figure, the place where cropping was applied should be marked (with a box), and molecular weight/size standards should be labeled wherever possible. Source Data files will be made available to reviewers during evaluation of revised manuscripts and, if your paper is eventually published in JCB, the files will be directly linked to specific figures in the published article.

The typical timeframe for revisions is three to four months. While most universities and institutes have reopened labs and allowed researchers to begin working at nearly pre-pandemic levels, we at JCB realize that the lingering effects of the COVID-19 pandemic may still be impacting some aspects of your work, including the acquisition of equipment and reagents. Therefore,

if you anticipate any difficulties in meeting this aforementioned revision time limit, please contact us and we can work with you to find an appropriate time frame for resubmission. Please note that papers are generally considered through only one revision cycle, so any revised manuscript will likely be either accepted or rejected.

Thank you for this interesting contribution to Journal of Cell Biology. You can contact us at the journal office with any questions at cellbio@rockefeller.edu.

Sincerely,

Elizabeth Miller, PhD
Monitoring Editor

Andrea L. Marat, PhD
Senior Scientific Editor

Journal of Cell Biology

Reviewer #1 (Comments to the Authors (Required)):

The study delves into the longstanding mysteries surrounding the function of adaptor protein 1 (AP-1) spanning over 35 years. To unravel the role of AP-1, the researchers devised innovative tools to explore the trafficking of AP-1 cargos and developed protocols for isolating AP-1 vesicles. These tools enabled them to investigate how, when, and where cargos are packed into AP-1 vesicles.

The investigation commenced with two AP-1 cargos, the cation-dependent mannose-6-phosphate receptor (CDMPR) and N-acetylglucosamine-1-phosphodiester alpha-N-acetylglucosaminidase (NAGPA), leaving the Golgi in separate vesicles. A non-AP-1 cargo, LAMP1, was used for comparison. The researchers confirmed that CD-MPR and NAGPA, in the presence of gadkin, colocalize with AP-1 in "peripheral foci," while LAMP1 does not. Surprisingly, they observed that NAGPA and LAMP1 use the same tubular carriers for transport, decorated with AP-1 punctae. CD-MPR, on the other hand, leaves the TGN in small vesicles.

In a subsequent analysis, the authors employed single vesicle microscopy to study vesicle fractions enriched with clathrin-coated vesicles (CCV) from cells carrying mRubytagged AP-1 and RUSH-membrane cargos. This allowed them to identify the percentage of AP-1-positive vesicles containing cargo, distinguishing between intracellular and endocytosed cargo using a nanobody. Results showed that after 2 hours of biotin addition, RUSH CD-MPR reached a steady state, with three-quarters of AP-1 vesicles containing CD-MPR, and 50% of these being endocytosed cargo.

In contrast, NAGPA plateaued after 40 minutes, with only a third of AP-1 vesicles containing NAGPA, and a higher percentage of endocytosed protein. This suggests that at least some of CDMPR and NAGPA are packaged into AP-1 vesicles shortly after leaving the TGN. Additionally, these findings indicated that NAGPA is directly transported to the cell surface, where it is endocytosed. Only 20% of AP-1 vesicles contained LAMP1. However, when co-trafficking of M6P-R and NAGPA was investigated, it was found that 80% of AP-1 vesicles containing NAGPA also had CD-MPR, supporting the original hypothesis.

To explore if AP-1 associates with tubular endosomes, the authors introduced internalized transferrin and analyzed the overlap with newly synthesized CD-MPR. The results demonstrated only a 20% overlap between CD-MPR and transferrin after 30 minutes in biotin, revealing the heterogeneity of AP-1 vesicles.

Further investigation into the collaboration between GGA2 and AP-1, suggested to facilitate lysosomal hydrolase targeting, revealed that while GGA and AP-1 move together, they are spatially separated. Their sequential action was ruled out, as no GGA spot that becomes AP-1 positive was observed.

Finally, the authors purified AP-1 vesicles and subjected them to SILAC-based mass spectrometry. Key findings included the presence of the AP-1 machinery, TGN resident proteins, late Golgi SNARES, peripheral membrane proteins (KIF13A), and the SM protein VPS45 and SPRYD7. Importantly, GGA2 was not detectable. The authors concluded that AP-1 retrieves proteins from post-Golgi compartments back to the TGN, akin to COPI in the early secretory pathway.

In conclusion, this paper resolves a longstanding question and greatly interests JCB readers. The authors' innovative techniques

contribute immensely to the trafficking community, and the study's rigor and comprehensiveness make it a noteworthy piece of work, requiring only minor revisions.

Comments to improve the text:

1. The introduction would benefit from shortening. The current version is very long.
2. Beautiful assay to discriminate TGN to the cell surface from endocytic vesicles. However, it would be constructive for the reader if explained in the introduction. How and why the CD-MPR traffics to the cell surface. Please also elaborate on the retrograde trafficking pathway. Does endocytosed CD-MPR end up in the same endosomes as the ones that recycle the unloaded CD-MPR complex?
3. For the role of AP-1 in recycling Golgi proteins, please include the following reference: Casler JC, Johnson N, Krahn AH, Pantazopoulou A, Day KJ, Glick BS. Clathrin adaptors mediate two sequential pathways of intra-Golgi recycling. *J Cell Biol.* 2022 Jan 3;221(1):e202103199. doi: 10.1083/jcb.202103199. Epub 2021 Nov 5. PMID: 34739034; PMCID: PMC8576872.
4. The text describing Figure 5 C-H is challenging to understand.

.....First, whereas CDMPR continued to increase in AP-1 vesicles between 40 minutes and 2 hours, NAGPA plateaued at 40 minutes. Second, after 2 hours in biotin, the majority of the AP-1 vesicles contained detectable CDMPR, while no more than a third of the AP-1 vesicles ever contained detectable NAGPA. Third, there was more endocytosed nanobody associated with the AP-1/NAGPA-positive vesicles than with AP-1/CDMPR-positive vesicles after 30 and 40 minutes in biotin. However, whereas nanobody uptake increased dramatically between 40 minutes and 2 hours in the SBP-GFP-CDMPR-expressing cells, this did not occur in the SBP-GFP-NAGPA-expressing cells.....

My suggestion: Firstly, while CDMPR levels in AP-1 vesicles continued to rise from 40 minutes to 2 hours, NAGPA reached a plateau at 40 minutes. Secondly, after 2 hours with biotin, most AP-1 vesicles contained detectable CDMPR, whereas no more than a third had detectable NAGPA. Thirdly, there was a higher association of endocytosed nanobody with AP-1/NAGPA-positive vesicles compared to AP-1/CDMPR-positive vesicles after 30 and 40 minutes in biotin. However, nanobody uptake significantly increased between 40 minutes and 2 hours in cells expressing SBP-GFP-CDMPR; no such increase occurred in cells expressing SBP-GFP-NAGPA.

5. The text describing Figure 6 is complex to comprehend.

.....Although their steady state distributions were distinct (Figure 6A; see also Figure 4C), and although they left the Golgi primarily in different carriers, ~80% of the AP-1 vesicles that contained NAGPA also contained CDMPR (Figure 6B and C and Supplemental Figure 3D).....

My suggestion: Despite having distinct steady-state distributions (as shown in Figure 6A and also observed in Figure 4C) and leaving the Golgi in different carriers, approximately 80% of AP-1 vesicles carrying NAGPA also included CDMPR (as depicted in Figure 6B and C, and Supplemental Figure 3D).

Experimental suggestion:

Does endocytosed CD-MPR end up in the same endosomes as the ones that recycle the unloaded CD-MPR complex? This would be important to know. The authors could perform immunostaining of RUSH-CD-MPR for 30 minutes and 2 hours and co-stain with endosomal markers.

Suggestions to improve the Figures:

Figure 1: You could add your new model here.

Figure 2: The cartoon in Figure 2A is challenging to understand. Please indicate with arrows where the newly synthesized proteins move to. Is the membrane structure above the Golgi an endosome? Is the circle the lysosome? How does the nanobody get from the cell surface to the TGN? Please include arrows with directions and add a Figure key.

Figure 3: A. Please also add the AP-1 image without channel merging. B. Please add the 90 minutes of biotin to the images.

Figure 4: It would be nice to include quantifying the events, i.e., how many NAGPA tubules emerging from the Golgi contain AP-1, etc.

Figure 8: It would be nice to show a zoom of one of the vesicles. Please also indicate that AP-1 captured the vesicles.

Reviewer #2 (Comments to the Authors (Required)):

Robinson et al. utilize elegant *in vivo* and biochemical experiments with cultured mammalian cells to dissect the functions of the AP-1 clathrin adaptor. This ancient and biologically critical protein complex has previously been implicated in various membrane traffic pathways. Most notably, there has been a long-standing assumption that AP-1 mediates anterograde transport of lysosomal hydrolases from the TGN to endosomes. The authors argue that in fact, mammalian AP-1 functions mainly (or perhaps exclusively) in the retrograde transport of late Golgi resident proteins from post-Golgi compartments back to the Golgi. Although their evidence in favor of retrograde transport is somewhat indirect, the data are shown to fit better to this model than to other models that have previously been considered. The view of AP-1 as an adaptor for retrograde transport fits with data from yeast and potentially also from plant cells. Therefore, the current study aims to reconcile decades' worth of results from multiple model systems to propose a unified model for AP-1 function across eukaryotes. This work will be of high interest to the cell biology community.

There was one major concern: the cargoes tested were all overexpressed using viral transduction. Overexpression could place a burden on the secretory pathway and saturate the membrane traffic machinery, thereby affecting the dynamics of cargo transit and possibly causing leakage into pathways that the cargoes do not normally follow. Did the authors see similar results using clones that expressed the cargoes at different levels? This issue is particularly relevant for NAGPA (the M6P uncovering enzyme), which travels to the plasma membrane in the experiments shown here. There is no physiological reason for NAGPA to leave the Golgi, so a likely interpretation is that NAGPA is leaking into secretory carriers due to overexpression. This point should be acknowledged and ideally addressed in some way.

Additional notes:

- 1) The significance of the gadkin colocalization experiment in Fig. 3B is not obvious. Is the goal to confirm that NAGPA and CDMPR are bona fide AP-1 cargoes? It's unclear that this analysis adds significantly to the story.
- 2) Despite LAMP-1 not being an AP-1 cargo, a substantial number of vesicles in Fig. 5H seem to be positive for both LAMP-1 and AP-1, almost as many as are positive for both NAGPA and AP-1 at 30 min after release from the ER. The authors describe this LAMP-1 signal as background, but it could reflect cargo mislocalization due to overexpression.
- 3) For the statement "CDMPR left the Golgi mainly in vesicular carriers, and we saw no obvious colocalization of these carriers with AP-1", data should either be shown or stated as data not shown.
- 4) Is it known that the GFP nanobody is only internalized after binding to a GFP-tagged surface protein? In other words, can the authors rule out that some of the nanobody is internalized by bulk endocytosis and then eventually encounters intracellular GFP-tagged proteins?
- 5) It is hard to extract much information from the videos. They should have time stamps, and they could also have arrows to mark structures of interest.
- 6) In Fig. 4A, the tubule containing NAGPA and LAMP1 at 7" still seems to be present at 21". Is that structure actually a transient tubular carrier, or is it a long-lived tubular projection?
- 7) The authors posit that the sensitivity of their system to far-red is low, but far-red-labeled transferrin was readily detected. Was the transferrin signal much stronger than that of overexpressed NAGPA-HALO or CDMPR-HALO?
- 8) There are several issues with Fig. 6C, which is a key experiment. (a) The text states that "~80% of the AP-1 vesicles that contained NAGPA also contained CDMPR". It took a while to understand this statement with reference to the figure. Presumably, the point is that the white portions of the bars are ~4x taller than the magenta portions. If so, this concept should be explained better. (b) Given the assertion that the microscopy system is less sensitive to far-red, it would make sense to repeat this experiment after swapping the HALO and GFP tags to see if the numbers change significantly. (c) In this experiment and in Fig. 5, it's unclear why the NAGPA levels in AP-1 vesicles are significantly lower than the CDMPR levels, especially given that NAGPA is well expressed. Additional discussion of this difference would be helpful.
- 9) For Fig. 6E, the experimental design and the interpretation are both unclear. Were the cells first fed labeled transferrin for an hour, or was transferrin added together with biotin? Is the loss of transferrin between 30 min and 2 h due to recycling of the internalized transferrin to the cell surface? More generally, why would internalized transferrin be expected to show up in AP-1 vesicles--is it a bulk marker?
- 10) It does not seem justified to state that "AP-1 vesicles can carry material that has been endocytosed, such as nanobodies and transferrin, supporting a role for AP-1 in trafficking out of endosomes". For example, although CDMPR bound to nanobody must have been endocytosed at some point, it could have recycled to the TGN before being captured in AP-1 vesicles.

11) It is unclear whether Chen et al. (2017) actually showed that the presence or absence of a luminal domain in an MPR will affect whether the protein partitions into tubules versus vesicles.

Reviewer #3 (Comments to the Authors (Required)):

Robinson et al. make a case for AP-1 acting to retrieve proteins back to the TGN, and that this "COPI-like" role is universal to eukaryotes. This is in contrast to the (wholly or partly) anterograde role usually ascribed to AP-1 in higher eukaryotes. The somewhat provocative results are significant, not only because of the first author's long-standing interest in AP-1, but also because the paper helps to resolve and contextualize the findings from many other groups. The authors also introduce a single vesicle analysis (SVA) approach which extends a microscopy approach first shown in Hirst et al., 2004 (PMID 15371541), to look at other vesicle components without the background noise of the cell. This approach will be very useful for analysis of other vesicle types. Overall, this is a very nice paper with an important message. I have only minor comments.

In the interests of reproducibility, it would be useful to make available the python script(s) used to do SVA.

I found Supplementary Figure S3 difficult to understand. As I see it, the (mean {plus minus} sd) percentages of the total number of spots are shown.

- 1) There is no y-axis label and the legend is unclear on what is plotted.
- 2) Is it showing the percentage of total spots? or AP-1 spots or something else? Some bars would stack to much greater than 100% which is not possible if it is the total population. So I got confused here.
- 3) The key is not precise. I understand that Red must be the total population of red spots, rather than those that are solely Red (and not also Green and/or Blue). The label R+G (for example) is unclear if it contains FR or not, i.e. are R+G in the white fraction included in the R+G fraction or not? Using set theory notation may help? White would be $R \cap G \cap FR$ and yellow would be $R \cap G \cap FR'$. Red could be defined as R or as $R \cap (G \cup FR)$ depending on what is being shown (I wasn't sure). This solution is possibly too nerdy for a cell biology paper but at least it is precise.
- 4) The figure is not colorblind-safe. I don't advocate for changing the colors because they have a meaning, but I think it would be possible to reorganise the bars on the x-axis and add labels so that colorblind folks can interpret too. For example, four groups instead of three, organised as red, green, blue (or just stack the blue on green and have two primary groups) and then show the "overlap" groups, including cyan; and add a label to say "overlap" underneath?

In Model 3, the origin of hydrolases and cargo that is being retrieved by AP-1 is mysterious because the GGA traffic is shown doing to a different compartment. Perhaps another (thin/dotted) arrow going from the GGA bud to the endosome compartment would be more accurate? I can understand that the authors don't want to draw additional compartments to keep in the theme of Model 1 and 2, but I guess that's another solution.

I like the direct style in which the manuscript is written. The narrative is strong and it's quite similar to listening to a seminar. Like a talk, if one is following along, then everything is fine, but I think that people less familiar with the topic could get lost. As an example, p. 13 "To address this question, we generated cells co-expressing SBP-GFP-CDMPR and SBP-HaloTag-NAGPA." It's not clear that this is in the background of streptavidin-KDEL hook and AP-1-mRuby2. If the thread is being followed then this is kind of obvious, but I'm pretty sure some students in my lab would not understand the subtlety.

In some other places, it wasn't quite clear what was meant, e.g. the significance of "There appeared to be even less of the ribosomal protein RPS17, which was usually undetectable in the bead samples." is not explained.

p. 8 "Several other Type I membrane proteins that had been identified as AP-1 cargo were also investigated..." the use of CDMPR, NAGPA and LAMP1 is justified, but the way this is written could be interpreted that the data with CPD, KIAA0319L, sortilin etc. are not shown for some other reason (cherry picking, data not shown). Perhaps saying that other proteins were screened (rather than investigated) and then justify why the cargos were chosen for investigation?

Margaret S. Robinson
Emeritus Professor of Molecular Cell Biology

Journal of Cell Biology
16 February, 2024

Dear Editors,

Thank you for the reviewers' comments on our manuscript, "The role of the AP-1 adaptor complex in outgoing and incoming membrane traffic" (202310071). We were very happy to hear that you and they agree that our paper "should be of high interest for the readership of *JCB*". We have now done a number of additional experiments and made modifications to the text, including some cuts to bring the character number down to <40,000. We have also generated Source Data files for our blots and gel. Our responses to the reviewers' comments are below.

Reviewer #1

Comments to improve the text:

1. The introduction would benefit from shortening. The current version is very long.

We realise that the introduction is longer than most other introductions in *JCB*, but we did want to include some of the history, and this goes back nearly 60 years. However, we have done some trimming, so the introduction is now slightly shorter.

2. Beautiful assay to discriminate TGN to the cell surface from endocytic vesicles. However, it would be constructive for the reader if explained in the introduction. How and why the CD-MPR traffics to the cell surface. Please also elaborate on the retrograde trafficking pathway. Does endocytosed CD-MPR end up in the same endosomes as the ones that recycle the unloaded CD-MPR complex?

Thank you! We opted not to include anything about the assay in the introduction. This is both because (as the reviewer points out) the introduction is already long, and because we wanted to focus it on AP-1, and not get into experimental protocols, which we felt belonged more in the results section. Regarding how and why the CDMPR traffics to the plasma membrane, as far as we are aware, this isn't entirely clear, because unlike the CIMPR, it doesn't bind hydrolases at pH 7.0-7.4. However, in the introduction, we do mention its trafficking to the cell surface as well as to different types of endosomes (page 4). The question of whether different pools of CDMPR end up in the same endosomes is an interesting one, and we answer it later in our response to the experimental suggestion.

3. For the role of AP-1 in recycling Golgi proteins, please include the following reference: Casler JC, Johnson N, Krahn AH, Pantazopoulou A, Day KJ, Glick BS. Clathrin adaptors mediate two sequential pathways of intra-Golgi recycling. *J Cell Biol.* 2022 Jan 3;221(1):e202103199. doi: 10.1083/jcb.202103199. Epub 2021 Nov 5. PMID: 34739034; PMCID: PMC8576872.

Thank you, we now include this reference (see page 23).

4. The text describing Figure 5 C-H is challenging to understand.

.....First, whereas CDMPR continued to increase in AP-1 vesicles between 40 minutes and 2 hours, NAGPA plateaued at 40 minutes. Second, after 2 hours in biotin, the majority of the AP-1 vesicles contained detectable CDMPR, while no more than a third of the AP-1 vesicles ever contained detectable NAGPA. Third, there was more endocytosed nanobody associated with the AP-1/NAGPA-positive vesicles than with AP-1/CDMPR-positive vesicles after 30 and 40 minutes in biotin. However,

whereas nanobody uptake increased dramatically between 40 minutes and 2 hours in the SBP-GFP-CDMMPR-expressing cells, this did not occur in the SBP-GFP-NAGPA-expressing cells.....

My suggestion: Firstly, while CDMMPR levels in AP-1 vesicles continued to rise from 40 minutes to 2 hours, NAGPA reached a plateau at 40 minutes. Secondly, after 2 hours with biotin, most AP-1 vesicles contained detectable CDMMPR, whereas no more than a third had detectable NAGPA. Thirdly, there was a higher association of endocytosed nanobody with AP-1/NAGPA-positive vesicles compared to AP-1/CDMMPR-positive vesicles after 30 and 40 minutes in biotin. However, nanobody uptake significantly increased between 40 minutes and 2 hours in cells expressing SBP-GFP-CDMMPR; no such increase occurred in cells expressing SBP-GFP-NAGPA.

Thank you, the rewrite is definitely clearer, and we have now incorporated it into the text (page 12).

5. The text describing Figure 6 is complex to comprehend. Although their steady state distributions were distinct (Figure 6A; see also Figure 4C), and although they left the Golgi primarily in different carriers, ~80% of the AP-1 vesicles that contained NAGPA also contained CDMMPR (Figure 6B and C and Supplemental Figure 3D).

My suggestion: Despite having distinct steady-state distributions (as shown in Figure 6A and also observed in Figure 4C) and leaving the Golgi in different carriers, approximately 80% of AP-1 vesicles carrying NAGPA also included CDMMPR (as depicted in Figure 6B and C, and Supplemental Figure 3D).

Again, we have incorporated this suggestion (page 13).

Experimental suggestion:

Does endocytosed CD-MPR end up in the same endosomes as the ones that recycle the unloaded CD-MPR complex? This would be important to know. The authors could perform immunostaining of RUSH-CD-MPR for 30 minutes and 2 hours and co-stain with endosomal markers.

We actually have a stably transduced cell population, called AHBMOM, which was not used in the original manuscript, but which allowed us to address this question, although not exactly in the way suggested by the referee. The AHBMOM cells co-express two different versions of CDMMPR: an SBP-HaloTag construct, and a BFP-tagged construct (i.e., no SBP, so the construct is always at steady state). Addition of mNeonGreen-tagged anti-GFP nanobody can be used to follow the fate of the BFP-tagged construct after endocytosis, because the nanobody recognises BFP but not its own tag. The BFP itself is only very weakly fluorescent, but as far as we can tell it mostly colocalises with the endocytosed nanobody.

The experiment is a bit complicated, and the figure we produced (shown on the following page) takes up a whole page, so we have included it for the referees to see without making it part of the actual paper. We hope this will be acceptable.

In the figure, panel E shows the basic design of the experiment. It's similar to other experiments in the paper, except that different versions of the same cargo protein, CDMMPR, are used to look at trafficking through the secretory and endocytic pathways; and, for the endocytic pathway, the BFP-tagged CDMMPR is invisible (shown in grey in the diagram in E). Panels A and B show widefield images of the cells, either without biotin or with biotin for 2 hours. In both cases, the cells were incubated with mNeonGreen-tagged nanobody for 1 hour. After 2 hours in biotin, most of the SBP-HaloTag-CDMMPR, which started out in the ER (A), now extensively colocalises with the endocytosed nanobody (B), showing that the two end up in the same endosomes. The images in panel C are stills from a video taken using Lattice SIM super-resolution microscopy, starting 28 minutes after biotin addition (see video for reviewers). Although at this stage most of the SBP-HaloTag-CDMMPR is still associated with the Golgi apparatus, there is also a small amount found in more peripheral structures containing endocytosed nanobody (some indicated with arrowheads). These structures are both tubular and round or irregular in shape, and some – especially the more tubular structures – appear to be positive for AP-1 as well.

No biotin

2 hr biotin

30 min biotin

Using single vesicle analysis (no biological repeats but 11 images for each time point), we find that after 2 hours in biotin, ~80% of the SBP-HaloTag-CDMPR in AP-1 vesicles colocalises with endocytosed nanobody, consistent with our images of whole cells. However, there is already colocalisation of the two markers in AP-1 vesicles after just 30 minutes in biotin, with ~60% of the AP-1 vesicles that contain SBP-HaloTag-CDMPR also containing endocytosed nanobody. Although this is higher than one would expect from the whole cell images, it is important to note that the pool of SBP-HaloTag-CDMPR in AP-1 vesicles has presumably already left the Golgi. The percentage of SBP-HaloTag-CDMPR that colocalises with endocytosed nanobody then increases with time.

Suggestions to improve the Figures:

Figure 1: You could add your new model here.

We did consider this, but in the end felt it would be better to keep the new model until the end. For one thing, this reflects our actual thought processes. This project took over 5 years, and for most of that time, we assumed that if AP-1 were entirely retrograde, the hydrolase receptors it recycled would be empty. We were surprised when we finally got the proteomic data showing that hydrolases as well as hydrolase receptors were enriched in the AP-1 vesicles. However, we then realised that AP-1 compartments wouldn't be sufficiently acidic for the hydrolases to dissociate, so that's how we wrote the manuscript, because we wanted the readers to follow our reasoning. We feel that putting a third model in at the beginning would jump the gun and possibly confuse readers. Thus, we would prefer to put just the first two models into Figure 1, and save the third model for the final figure.

Figure 2: The cartoon in Figure 2A is challenging to understand. Please indicate with errors where the newly synthesized proteins move to. Is the membrane structure above the Golgi an endosome? Is the circle the lysosome? How does the nanobody get from the cell surface to the TGN? Please include arrows with directions and add a Figure key.

We take the referee's point that Figure 2A could be clearer. However, we made a conscious decision not to label the organelles because this then gets into the question of what is an early endosome, what is a recycling endosome, etc. We've added more arrows, which we hope will make the diagram clearer, and we've expanded the figure legend to include more explanation.

Figure 3: A. Please also add the AP-1 image without channel merging. B. Please add the 90 minutes of biotin to the images.

We have followed both of these suggestions (see the new version of Figure 3).

Figure 4: It would be nice to include quantifying the events, i.e., how many NAGPA tubules emerging from the Golgi contain AP-1, etc.

We agree that it would be nice to quantify some of the events in the live imaging. However, we feel that our single vesicle analysis is an even more quantitative way of addressing questions such as when NAGPA starts to be incorporated into AP-1 vesicles, because it is based on thousands of individual puncta.

Figure 8: It would be nice to show a zoom of one of the vesicles. Please also indicate that AP-1 captured the vesicles.

We have followed both of these suggestions.

Reviewer #2

There was one major concern: the cargoes tested were all overexpressed using viral transduction. Overexpression could place a burden on the secretory pathway and saturate the membrane traffic

machinery, thereby affecting the dynamics of cargo transit and possibly causing leakage into pathways that the cargoes do not normally follow. Did the authors see similar results using clones that expressed the cargoes at different levels? This issue is particularly relevant for NAGPA (the M6P uncovering enzyme), which travels to the plasma membrane in the experiments shown here. There is no physiological reason for NAGPA to leave the Golgi, so a likely interpretation is that NAGPA is leaking into secretory carriers due to overexpression. This point should be acknowledged and ideally addressed in some way.

We completely agree that overexpression is a concern, even though we were careful to use a moderate promoter. Allowing membrane proteins to accumulate in the ER and then releasing them could also put a burden on the secretory pathway, even with endogenous levels of expression. We searched for an antibody against CDMPR early on in this project, to compare expression levels in wild-type and transduced cells, but unfortunately there do not seem to be any antibodies available that work for Western blotting. NAGPA is the least well characterised of the three membrane proteins we studied, and although there are antibodies one can buy, none of these have been published, and it has been our experience that many – perhaps most – commercial antibodies just don't work (when we have tried to make our own antibodies against conserved proteins, at least half of them were no good). Therefore, we decided to focus on LAMP1, as it is well characterised and a number of antibodies have been used successfully, including by some of our colleagues.

We tried two LAMP1 antibodies from Abcam initially, a rabbit polyclonal antiserum, 24170, which had been used by Paul Luzio's lab for immunolocalisation studies and was reported on the website to work for Western blotting as well, and a mouse monoclonal antibody, H4A3, used by David Gershlick's lab and also reported to work for both immunolocalisation and Western blotting. We tested both antibodies first on homogenates of cells that had been incubated with biotin for 40 minutes. The Western blots below, showing the same samples labelled with each of the two antibodies, demonstrate that both antibodies appear to work. However, the signal-to-background for 24170 was too weak to resolve the endogenous wild-type LAMP1, while H4A3 only seems to see the endogenous LAMP1 and not the tagged construct (we had to use it more concentrated than recommended, but the mobility and fuzziness of the ~120K bands look right). The only information we could find about its epitope was that it recognises the luminal domain of LAMP1, so we suspect that the luminal SBP and/or GFP tags interfered with antibody binding. We then tried another commercial monoclonal anti-LAMP1, H5G11 from Santa Cruz. This one also looked all right on the website, but when we tried it at both the recommended dilution and three times more concentrated, we got no signal. So we then tried it very concentrated indeed (diluted 1:5) and saw a lot of bands, but we suspect these are all non-specific.

In our paper, we show that SBP-GFP-CDMPR is enriched in CCV-enriched fractions (see Figure 5G), so we then probed Western blots of both cell homogenates and CCV-enriched fractions with antibodies 24170 and H4A3, to see if we could compare the enrichment of endogenous and tagged LAMP1.

Consistent with the data in Figure 5G, the higher molecular weight band labelled with antibody 24170 was enriched in the prep. However, we couldn't see anything that looked like endogenous CDMPR in the blot labelled with 24170, and we also weren't convinced that we could see anything specific in the blot labelled with H4A3 (the bands labelled with an asterisk are clathrin heavy chain). Thus, all we can say is that the LAMP1 construct is definitely overexpressed, because of the results on wild-type and SBP-GFP-LAMP1-expressing cells labelled with 24170. Our results also suggest that endogenous LAMP1 may not be enriched in our CCV-enriched fraction, or at least not as much as the tagged protein. But we cannot quantify how much the protein is overexpressed because of the lack of a suitable antibody. In the original version of our manuscript, we were careful not to imply that any of the tagged membrane proteins (CDMPR, NAGPA, and LAMP1) were expressed at endogenous levels. However, we now come right out and say that the membrane proteins are likely to be overexpressed (page 8),

To answer the question about clones that express cargoes at different levels, we didn't actually make clonal cell lines except for the *AP1G1* knockout, but instead used stably transduced populations of cells. Thus, we haven't been able to compare different clones. However, even though (as the reviewer points out) Golgi-localised proteins such as NAGPA have no physiological reason to leave the Golgi, it has long been known that they do, and that they then come back again (e.g., see Chapman and Munro, 1994). As we show in Figure 3A, the steady state localisation of the NAGPA construct is still primarily at the TGN, which means that in spite of overexpression and increased exit from the Golgi, the cell is still able to cope and to keep most of the NAGPA in the right compartment. However, we now include a caveat in the text to say that overexpressed NAGPA may be leaking into the LAMP1-containing tubules (page 10). One potential concern is that excessive amounts of NAGPA in the tubules might promote ectopic recruitment of AP-1. To address this possibility, we carried out live imaging on cells expressing just SBP-GFP-LAMP1, and again saw AP-1 puncta on LAMP1-positive tubules leaving the Golgi. This indicates that AP-1 is recruited onto these tubules whether or not the cells are overexpressing an AP-1 cargo protein. Stills from our videos are shown below, and the data can be found in our revised manuscript in Supplemental Figure S1C and in Video 4.

Additional notes:

1) The significance of the *gadkin* colocalization experiment in Fig. 3B is not obvious. Is the goal to confirm that NAGPA and CDMPR are bona fide AP-1 cargoes? It's unclear that this analysis adds significantly to the story.

The reviewer is correct, we did this experiment to confirm that NAGPA and CDMPR are bona fide AP-1 cargoes. We think this is important in light of the rather subtle difference between NAGPA and LAMP1 in the single vesicle analysis experiments (see below).

2) Despite LAMP-1 not being an AP-1 cargo, a substantial number of vesicles in Fig. 5H seem to be positive for both LAMP-1 and AP-1, almost as many as are positive for both NAGPA and AP-1 at 30

min after release from the ER. The authors describe this LAMP-1 signal as background, but it could reflect cargo mislocalization due to overexpression.

We agree with the reviewer that some of the LAMP1 may be spilling into AP-1 vesicles because of overexpression, and we now acknowledge this possibility (page 13).

3) For the statement "CDMPR left the Golgi mainly in vesicular carriers, and we saw no obvious colocalization of these carriers with AP-1", data should either be shown or stated as data not shown.

We now state that this is data not shown (page 9).

4) Is it known that the GFP nanobody is only internalized after binding to a GFP-tagged surface protein? In other words, can the authors rule out that some of the nanobody is internalized by bulk endocytosis and then eventually encounters intracellular GFP-tagged proteins?

Although trace amounts of nanobody are likely to enter the cell by bulk endocytosis, we carefully titrated the nanobody to minimise non-specific uptake. In the Western blots shown below, cells expressing SBP-GFP-CDMPR were incubated either with or without biotin, and given different concentrations of nanobody for the final 40 minutes. Only the biotin-treated cells were expected to take up nanobody specifically, because in the untreated cells, the construct should remain in the ER. The data show that at 9 $\mu\text{g/ml}$, there is very little uptake of nanobody in the cells that were not given biotin. The trace amounts that are seen could be due to bulk endocytosis, to a small fraction of cells that do not express sufficient Hook, or to a combination of both. Comparison of the no biotin vs 2 hr biotin lanes indicates that $\sim 2\%$ of the uptake at 9 $\mu\text{g/ml}$ nanobody may be non-specific, vs $\sim 75\%$ of the uptake at 72 $\mu\text{g/ml}$. In all of the experiments shown in the paper, we used the nanobody at concentrations of 5-10 $\mu\text{g/ml}$, which is why we are confident that (most of) the nanobody was only internalised after binding to a GFP-tagged surface protein. Although this figure is for the referees only and does not appear in the paper, we now mention in the Materials and Methods that the nanobody was titrated to find the optimum concentration (page 27).

5) It is hard to extract much information from the videos. They should have time stamps, and they could also have arrows to mark structures of interest.

We have now modified the videos to include both time stamps and arrows.

6) In Fig. 4A, the tubule containing NAGPA and LAMP1 at 7" still seems to be present at 21". Is that structure actually a transient tubular carrier, or is it a long-lived tubular projection?

It could certainly be a long-lived tubular projection (i.e., present for at least 14 seconds), but these tubular projections then give rise to tubular carriers, and these carriers are not all that transient, because they can persist for 30 seconds or more (see Pereira et al., *J. Cell Biol.* 2023).

7) The authors posit that the sensitivity of their system to far-red is low, but far-red-labeled transferrin was readily detected. Was the transferrin signal much stronger than that of overexpressed NAGPA-HALO or CDMPR-HALO?

The far-red transferrin signal appeared to be similar to that of HaloTag-NAGPA and HaloTag-CDMPR, and both far-red transferrin and HaloTag-CDMPR at steady state filled about a third of the vesicles after 2 hours in biotin. However, GFP-CDMPR at steady state filled over two-thirds of the vesicles. The main reason we suggested that our microscope had relatively low far-red sensitivity was that the beads we used as a fiducial marker, which fluoresce in four different wavelengths, had a much lower signal in far-red than in green, red, or blue, and we now explain this (page 12).

However, we didn't know for sure whether this was the only reason we never detected as much far-red cargo as green cargo in our AP-1-positive vesicles, and we wanted to make a direct comparison. Therefore, we used an antibody against streptavidin-binding peptide (SBP), which is present in all of our cargo constructs, to probe blots of some of our cells that had been treated with biotin for 40 minutes. To our surprise, we found that the GFP-tagged constructs were actually expressed at ~3 times higher levels than the same constructs with HaloTag. The promoters were identical; the only differences between the constructs were their tags, but GFP-tagged membrane proteins are inherently more highly expressed. Our new data, shown on the left, are now included in Supplemental Figure S4A and described on page 13. The green and blue circles are GFP- and HaloTag-CDMPR, respectively, and the green and blue squares are GFP- and HaloTag-NAGPA, respectively.

8) There are several issues with Fig. 6C, which is a key experiment. (a) The text states that "~80% of the AP-1 vesicles that contained NAGPA also contained CDMPR". It took a while to understand this statement with reference to the figure. Presumably, the point is that the white portions of the bars are ~4x taller than the magenta portions. If so, this concept should be explained better. (b) Given the assertion that the microscopy system is less sensitive to far-red, it would make sense to repeat this experiment after swapping the HALO and GFP tags to see if the numbers change significantly. (c) In this experiment and in Fig. 5, it's unclear why the NAGPA levels in AP-1 vesicles are significantly lower than the CDMPR levels, especially given that NAGPA is well expressed. Additional discussion of this difference would be helpful.

We realise that Figure 6C is somewhat confusing, and we have now attempted to clarify it by including the reviewer's comment about white and magenta portions of the bars in the figure legend. Swapping the tags does make sense, but it would mean creating an additional cell type, and then carrying out three biological repeats of the SVA experiment, which would take several months. We feel that the experiment we include, with cells coexpressing SBP-GFP-CDMPR and SPP-HaloTag-CDMPR, provides adequate support for our claim, and as described above, we now also include a Western blot to compare the expression levels of GFP and HaloTag on the same constructs. To answer the question about why NAGPA levels in AP-1 vesicles are lower than CDMPR levels, when there is at least as much expression of NAGPA as of CDMPR, we hypothesise that this is a reflection of the distribution of the proteins in the cell. We made this point in our original manuscript, but we now spell it out more clearly (pages 14 and 20).

9) For Fig. 6E, the experimental design and the interpretation are both unclear. Were the cells first fed labeled transferrin for an hour, or was transferrin added together with biotin? Is the loss of transferrin between 30 min and 2 h due to recycling of the internalized transferrin to the cell surface? More

generally, why would internalized transferrin be expected to show up in AP-1 vesicles--is it a bulk marker?

We have expanded the figure legend and Materials and Methods (pages 29-30) to make the experimental design clearer. When doing a CCV prep, we normally prepare 12 confluent 150-cm dishes, and use 3 dishes for each of the 4 time points. We stagger the time points so that we process one set of three dishes every 20 minutes. This is because it takes some time to rinse the plates, scrape the cells and rinse and scrape again, do the homogenisation, and then wash the homogenising and scraping equipment in time for the next set of dishes. So, depending on the length of time in biotin, the transferrin was added either before or after the biotin, and the dishes were washed and processed after 1 hour in transferrin. In other words, all of the cells were given transferrin for exactly one hour, and then processed immediately, so one would expect equal amounts of recycling in all the cells.

To answer the question of why internalised transferrin would show up in AP-1 vesicles, as we point out in the text (page 14), there are EM studies going back to 1998 showing that there is a lot of AP-1 on tubular endosomes containing transferrin, and we show AP-1 puncta on structures containing endocytosed transferrin in Videos 5 and 6. In addition, our vesicle capture experiments show that although the transferrin receptor is not efficiently captured by anti-mRuby2 (Table 1), it is so abundant that there is actually quite a lot of it in the captured vesicles (Table 2). Moreover, we have previously shown that AP-1 knockdowns cause a subtle but highly reproducible inhibition of transferrin recycling (Hirst et al., 2005, PubMed 15758025). But whether AP-1 plays an active role in the transferrin pathway, or whether the knockdown effects are indirect, and whether transferrin and its receptor are actually sorted by AP-1 or just spill into AP-1 vesicles because they're so abundant, are still unclear.

Regarding the loss of transferrin between 30 minutes and 2 hours, this shouldn't really be happening, but we can't deny that it does, even though, as we say in the text, roughly 30% of the AP-1 vesicles contain transferrin at all time points. We have extracted the relevant parts of the bar graphs from Supplemental Figure S3F and show them below. It is clear that there is quite a lot of variability between experiments, with particularly strong differences between time points in experiment 1, and there is even more variability in internalised transferrin in puncta that do not contain AP-1 (see the bars on the right). We suspect there may be a fairly trivial explanation for this, like how often and for how long the incubator door was opened during the transferrin incubation.

10) It does not seem justified to state that "AP-1 vesicles can carry material that has been endocytosed, such as nanobodies and transferrin, supporting a role for AP-1 in trafficking out of endosomes". For example, although CDMPR bound to nanobody must have been endocytosed at some point, it could have recycled to the TGN before being captured in AP-1 vesicles.

We thank the reviewer for pointing this out. We agree, and have now rewritten that part (page 15).

11) It is unclear whether Chen et al. (2017) actually showed that the presence or absence of a luminal domain in an MPR will affect whether the protein partitions into tubules versus vesicles.

The reviewer is correct, Chen et al. never showed whether it was the luminal domain, the transmembrane domain, or both that determined whether an MPR left the Golgi in tubules or vesicles. This is why we wrote “the luminal and/or transmembrane domains”. However, if it is indeed the luminal domain, that could explain why Waguri et al. saw MPRs exiting the Golgi in tubules while we did not.

Reviewer #3

In the interests of reproducibility, it would be useful to make available the python script(s) used to do SVA.

This was always our intention, and the script is now available at <https://github.com/GershlickLab/SVA/> (see page 31).

I found Supplementary Figure S3 difficult to understand. As I see it, the (mean {plus minus} sd) percentages of the total number of spots are shown.

1) There is no y-axis label and the legend is unclear on what is plotted.

2) Is it showing the percentage of total spots? or AP-1 spots or something else? Some bars would stack to much greater than 100% which is not possible if it is the total population. So I got confused here.

3) The key is not precise. I understand that Red must be the total population of red spots, rather than those that are solely Red (and not also Green and/or Blue). The label R+G (for example) is unclear if it contains FR or not, i.e. are R+G in the white fraction included in the R+G fraction or not? Using set theory notation may help? White would be $R \cap G \cap \overline{FR}$ and yellow would be $R \cap \overline{G} \cap \overline{FR}$. Red could be defined as R or as $R \cap (G \cup FR)$ depending on what is being shown (I wasn't sure). This solution is possibly too nerdy for a cell biology paper but at least it is precise.

4) The figure is not colorblind-safe. I don't advocate for changing the colors because they have a meaning, but I think it would be possible to reorganise the bars on the x-axis and add labels so that colorblind folks can interpret too. For example, four groups instead of three, organised as red, green, blue (or just stack the blue on green and have two primary groups) and then show the "overlap" groups, including cyan; and add a label to say "overlap" underneath?

We agree that Supplementary Figure S3 is somewhat confusing, and we have tried to make it clearer.

Starting with the reviewer's third comment, the colours in the figure reflect the actual colours we saw in our images, using the three primary colours of light for the three channels: red, green, and blue (far-red was coloured blue). Thus, the red bars show the percentage of spots that were red only (i.e., containing only AP-1 with no other colours detectable), the green bars show green only, and the blue bars show far-red only. We have added “only” to the figure and hope this makes it clearer.

Similarly, the R+G bars show spots that were only red and green, and so on; the colours are those perceived by the non-colourblind human eye when the primary colours of light are mixed (see left). Thus, if a spot contains both red AP-1 and green CDMPR, it appears yellow, while if it contains blue nanobody as well, it appears white. We have expanded the figure legend to try to explain this better.

Moving on to the first point, we have now labelled the y-axes. They are (like the y-axes in Figures 5 and 6) the percentage of AP-1 vesicles. However, because we also show vesicles that don't contain AP-1, the percentage is normalised for those points. This explains why the percentage of some of the non-AP-1-containing vesicles is >100%.

We hope that these additions to the figure and figure legend have clarified points 1-3. Regarding point 4, we were concerned about whether the figure would be colourblind-safe, so we showed it to two colleagues with X-linked red-green colourblindness. Both were able to match the chart in the upper left

with the actual bars. We aren't so keen to move the bars around because they were arranged in this particular order for a reason. The first set of 4 bars shows spots that contain AP-1 plus one or more other labels, and these are the bars that were used to generate Figures 5 and 6. The middle set shows spots that contain AP-1 only; and the final set shows spots that contain only the other labels, without any detectable AP-1. We have added this information to the figure and hope that it is now clear enough to be interpreted by all readers.

In Model 3, the origin of hydrolases and cargo that is being retrieved by AP-1 is mysterious because the GGA traffic is shown doing to a different compartment. Perhaps another (thin/dotted) arrow going from the GGA bud to the endosome compartment would be more accurate? I can understand that the authors don't want to draw additional compartments to keep in the theme of Model 1 and 2, but I guess that's another solution.

This is a good idea, and we have added an arrow with a dotted line to show that the GGA compartment matures into the AP-1 compartment, and have also expanded the figure legend.

I like the direct style in which the manuscript is written. The narrative is strong and it's quite similar to listening to a seminar. Like a talk, if one is following along, then everything is fine, but I think that people less familiar with the topic could get lost. As an example, p. 13 "To address this question, we generated cells co-expressing SBP-GFP-CDMPR and SBP- HaloTag-NAGPA." It's not clear that this is in the background of streptavidin-KDEL hook and AP-1-mRuby2. If the thread is being followed then this is kind of obvious, but I'm pretty sure some students in my lab would not understand the subtlety. In some other places, it wasn't quite clear what was meant, e.g. the significance of "There appeared to be even less of the ribosomal protein RPS17, which was usually undetectable in the bead samples." is not explained.

We can see that more clarity is needed, and we have rewritten these sections (pages 13 and 16).

p. 8 "Several other Type I membrane proteins that had been identified as AP-1 cargo were also investigated..." the use of CDMPR, NAGPA and LAMP1 is justified, but the way this is written could be interpreted that the data with CPD, KIAA0319L, sortilin etc. are not shown for some other reason (cherry picking, data not shown). Perhaps saying that other proteins were screened (rather than investigated) and then justify why the cargos were chosen for investigation?

Thank you, we have incorporated this suggestion (page 8).

We thank all three referees for their constructive suggestions, which we feel have improved our manuscript. We hope that with the changes we have made, the manuscript will now be acceptable for publication in *JCB*. Thank you again for considering our manuscript, and we look forward to hearing from you.

Yours sincerely,

Margaret S. Robinson

Cambridge Institute for Medical Research
Keith Peters Building
Cambridge CB2 0XY
Email: msr12@cam.ac.uk
www.cimr.cam.ac.uk

February 20, 2024

RE: JCB Manuscript #202310071R

Prof. Margaret S. Robinson
University of Cambridge
CIMR
Cambridge CB2 0XY
United Kingdom

Dear Prof. Robinson:

Thank you for submitting your revised manuscript entitled "The role of the AP-1 adaptor complex in outgoing and incoming membrane traffic". We agree you have addressed all essential concerns and would be happy to publish your paper in JCB pending final revisions necessary to meet our formatting guidelines (see details below). In your final revision please include your rebuttal figure to reviewer 1 as a supplemental figure. We can permit an extension to the SI figure count if necessary.

A. MANUSCRIPT ORGANIZATION AND FORMATTING:

- 1) Text limits: Character count for Articles is < 40,000, not including spaces. Count includes abstract, introduction, results, discussion, and acknowledgments. Count does not include title page, figure legends, materials and methods, references, tables, or supplemental legends.
- 2) Figures limits: Articles may have up to 10 main text figures.
- 3) Figure formatting: Scale bars must be present on all microscopy images, * including inset magnifications. Molecular weight or nucleic acid size markers must be included on all gel electrophoresis.
- 4) Statistical analysis: Error bars on graphic representations of numerical data must be clearly described in the figure legend. The number of independent data points (n) represented in a graph must be indicated in the legend. Statistical methods should be explained in full in the materials and methods. For figures presenting pooled data the statistical measure should be defined in the figure legends. Please also be sure to indicate the statistical tests used in each of your experiments (either in the figure legend itself or in a separate methods section) as well as the parameters of the test (for example, if you ran a t-test, please indicate if it was one- or two-sided, etc.). Also, if you used parametric tests, please indicate if the data distribution was tested for normality (and if so, how). If not, you must state something to the effect that "Data distribution was assumed to be normal but this was not formally tested."
- 5) Abstract and title: The abstract should be no longer than 160 words and should communicate the significance of the paper for a general audience. The title should be less than 100 characters including spaces. Make the title concise but accessible to a general readership.
- 6) Materials and methods: Should be comprehensive and not simply reference a previous publication for details on how an experiment was performed. Please provide full descriptions in the text for readers who may not have access to referenced manuscripts.
- 7) **** All antibodies, cell lines, animals, and tools used in the manuscript should be described in full, including accession numbers for materials available in a public repository such as the Resource Identification Portal. Please be sure to provide the sequences for all of your primers/oligos and RNAi constructs in the materials and methods. You must also indicate in the methods the source, species, and catalog numbers (where appropriate) for all of your antibodies. Please also indicate the acquisition and quantification methods for immunoblotting/western blots. ****
- 8) Microscope image acquisition: The following information must be provided about the acquisition and processing of images:
 - a. Make and model of microscope
 - b. Type, magnification, and numerical aperture of the objective lenses
 - c. Temperature
 - d. Imaging medium

- e. Fluorochromes
- f. Camera make and model
- g. Acquisition software
- h. Any software used for image processing subsequent to data acquisition. Please include details and types of operations involved (e.g., type of deconvolution, 3D reconstitutions, surface or volume rendering, gamma adjustments, etc.).

10) Supplemental materials: There are strict limits on the allowable amount of supplemental data. Articles may typically have up to 5 supplemental figures, we can allow 6 to include the addition requested figure. Please be sure to correct the callouts in the text to reflect any changes. Please also note that tables, like figures, should be provided as individual, editable files. A summary of all supplemental material should appear at the end of the Materials and methods section.

13) ORCID IDs: ORCID IDs are unique identifiers allowing researchers to create a record of their various scholarly contributions in a single place. Please note that ORCID IDs are now *required* for all authors. At resubmission of your final files, please be sure to provide your ORCID ID and those of all co-authors.

Please note that JCB now requires authors to submit Source Data used to generate figures containing gels and Western blots with all revised manuscripts. This Source Data consists of fully uncropped and unprocessed images for each gel/blot displayed in the main and supplemental figures. Since your paper includes cropped gel and/or blot images, please be sure to provide one Source Data file for each figure that contains gels and/or blots along with your revised manuscript files. File names for Source Data figures should be alphanumeric without any spaces or special characters (i.e., SourceDataF#, where F# refers to the associated main figure number or SourceDataFS# for those associated with Supplementary figures). The lanes of the gels/blots should be labeled as they are in the associated figure, the place where cropping was applied should be marked (with a box), and molecular weight/size standards should be labeled wherever possible.

Journal of Cell Biology now requires a data availability statement for all research article submissions. These statements will be published in the article directly above the Acknowledgments. The statement should address all data underlying the research presented in the manuscript. Please visit the JCB instructions for authors for guidelines and examples of statements at (<https://rupress.org/jcb/pages/editorial-policies#data-availability-statement>).

B. FINAL FILES:

****It is JCB policy that if requested, original data images must be made available to the editors. Failure to provide original images upon request will result in unavoidable delays in publication. Please ensure that you have access to all original data images prior to final submission.****

****The license to publish form must be signed before your manuscript can be sent to production. A link to the electronic license to publish form will be sent to the corresponding author only. Please take a moment to check your funder requirements before choosing the appropriate license.****

Thank you for this interesting contribution, we look forward to publishing your paper in Journal of Cell Biology.

Sincerely,

Elizabeth Miller, PhD
Monitoring Editor

Andrea L. Marat, PhD
Senior Scientific Editor

Journal of Cell Biology